# Rethinking Addressing in Language Models via Contextualized Equivariant Positional Encoding

**Jiajun Zhu** [* 1 2]  **Peihao Wang** [* 1]  **Ruisi Cai** [1]  **Jason D. Lee** [3]  **Pan Li** [4]  **Zhangyang Wang** [1]

 https://github.com/VITA-Group/TAPE

## Abstract

Transformers rely on both content-based and position-based addressing mechanisms to make predictions, but existing positional encoding techniques often diminish the effectiveness of position-based addressing. Many current methods enforce rigid patterns in attention maps, limiting the ability to model long-range dependencies and adapt to diverse tasks. Additionally, most positional encodings are learned as general biases, lacking the specialization required for different instances within a dataset. To address this, we propose conTextualized equivariAnt Position Encoding (**TAPE**), a novel framework that enhances positional embeddings by incorporating sequence content across layers. TAPE introduces dynamic, context-aware positional encodings, overcoming the constraints of traditional fixed patterns. We show that TAPE can provably facilitate LLM reasoning ability by emulating a broader class of algorithms. By enforcing permutation and orthogonal equivariance, TAPE ensures the stability of positional encodings during updates, improving long-context ability. Our method can be easily integrated into pre-trained transformers, offering parameter-efficient fine-tuning with minimal overhead. Extensive experiments show that TAPE achieves superior performance in language modeling, arithmetic reasoning, and long-context retrieval tasks compared to existing positional embedding techniques.

## 1. Introduction

Attention mechanisms are a core component of many modern deep learning architectures, enabling models to selectively focus on relevant information within a given context. Transformer models (Vaswani et al., 2017) and their numerous variants (Carion et al., 2020; Dosovitskiy et al., 2021; Zhao et al., 2021), which are fundamentally driven by attention, have revolutionized tasks involving sequential and spatial data, such as text (Kitaev et al., 2020), image (Dosovitskiy et al., 2021), and point cloud (Zhao et al., 2021). More recently, large transformer models have become dominant in natural language understanding, language generation, and complex reasoning (Brown et al., 2020).

Delving into attention's underlying paradigm, the prediction made for each token is expressed as a weighted aggregation over the representations of other tokens. Due to the softmax function, attention often generates a sparse mask, extracting a limited subset of tokens for interaction. Through this interpretation, attention can be understood as an *addressing* mechanism (Hopfield, 1982; Pagiamtzis & Sheikholeslami, 2006) that searches the context, locating and retrieving token representations deemed most relevant or important.

Since the attention score is computed upon token features and positions (see Sec. 2), transformers' addressing ability can be further decomposed into two fundamental mechanisms: *content-based* addressing and *position-based* addressing. Content-based addressing recognizes relevant tokens through feature similarity, while position-based addressing is facilitated by positional encoding techniques, designed to (ideally) enable random access along the sequence via indexing. It is important to let the two mechanisms cooperate to tackle more complex tasks, such as in-context retrieval (Hinton & Anderson, 2014; Ba et al., 2016), arithmetic (Lee et al., 2023; McLeish et al., 2024b), counting (Golovneva et al., 2024), logical computation (Liu et al., 2024), and reasoning (Wei et al., 2022; Rajani et al., 2019; Dziri et al., 2024). However, we contend that the role of position-based addressing is limited, if not diminishing, in modern transformer architectures (Ebrahimi et al., 2024).

---

[*]Equal contribution. Work was partially done when J. Zhu was an undergraduate at ZJU. [1]University of Texas at Austin [2]Zhejiang University [3]Princeton University [4]Georgia Tech. Correspondence to: Jiajun Zhu <jiajunzhu@utexas.edu>.

*Proceedings of the 42$^{nd}$ International Conference on Machine Learning*, Vancouver, Canada. PMLR 267, 2025. Copyright 2025 by the author(s).

It has not escaped our notice that most existing positional encodings weakens the position-based addressing capability. Recent works (Press et al., 2021b; Su et al., 2024; Chi et al., 2022b; Sun et al., 2022) impose a fixed and somewhat artisanal pattern on attention maps, typically adopting a decaying pattern in relation to relative distances, thereby enforcing a locality bias. This rigidity limits the ability of attention to model long-range dependencies and operate on a flexible group of tokens. Although some positional encodings have trainable parameters (Vaswani et al., 2017; Shaw et al., 2018; Chi et al., 2022a; Li et al., 2023), the hypothesis space is often excessively constrained.

Perhaps more crucially, most existing positional encodings are handcrafted and learned as a general bias across the entire dataset, lacking specialization and adaptability to specific instances informed by the context. In more advanced tasks, such as general-purpose algorithmic reasoning, LLMs rely on attention to internalize the representation of a computation graph, which must vary according to the problem specified in the context (Elhage et al., 2021; Ameisen et al., 2025). However, transformers without context-aware position-based addressing have been theoretically shown to represent only a restricted family of algorithms (Merrill & Sabharwal, 2023).

To unleash the power of position-based addressing, we endeavor to design a more universal and generic position encoding for language transformers. We introduce *Contextualized Equivariant Positional Encoding* (**TAPE**), a novel framework designed to contextualize positional embeddings by incorporating sequence content. TAPE continually progresses information flow between positional embeddings and token features via specialized attention and MLP layers. To ensure the stability during model updates, we enforce permutation and orthogonal group equivariance properties on attention and MLP layers. This approach is inspired from the studies for geometric deep learning which represents structural patterns (e.g., graphs) by integrating token features with their geometric properties while preserving inherent physical symmetries (Wang et al., 2024b; Huang et al., 2024). Building on this motivation, we encode algorithmic structures into transformers and theoretically show that TAPE achieves greater representational power, capturing algorithms of higher complexity than those representable by standard transformers. Further on, by enforcing equivariance priniples, TAPE maintains the relative relationships between encoded positions, ensuring transformer's long-context ability and length generalization.

Technically, we extend conventional vectorized positional embeddings into a multi-dimensional tensor, which enriches interactions between positional embeddings and token features. In the attention mechanism, TAPE incorporates the pairwise inner product between positional encodings, allowing attention values to be computed based on not only token similarities but also positional proximities. We additionally customize an MLP layer that directly mixes token features with positional encodings, while preserving orthogonal equivariance.

We demonstrate the superior performance of TAPE on arithmetic reasoning tasks (McLeish et al., 2024a), which require LLMs to effectively locate/address and retrieve specific tokens, as well as on representative natural language tasks, including SCROLLS (Shaham et al., 2022) and passkey retrieval (Mohtashami & Jaggi, 2023), to validate the generalizability of the framework.

Our contributions are summarized as follows:

- We introduce TAPE, a novel framework learning to represent token positions in the feature space jointly with sequential learning. TAPE contextualizes positional embeddings with sequence content across layers to enhance the position-addressing ability of transformers. We further enforce TAPE with permutation and orthogonal equivariance to guarantee the stability of positional encodings during the update. We theoretically show that TAPE achieves higher expressivness for algorithmic reasoning.

- We propose practical implementations for our TAPE, which extends conventional positional embeddings into multi-dimensional and facilitates attention and MLP in transformers with two levels of equivariance. We also show that TAPE has hardware-efficient implementation and can be used as a drop-in component into extant pre-trained models for parameter-efficient fine-tuning.

- Extensive experiments showcase TAPE's superiority in both training from scratch and parameter-efficient fine-tuning scenarios for language modeling as well as downstream tasks such as arithmetic reasoning and long-context retrieval. TAPE achieves state-of-the-art performance in language modeling, surpassing baselines in perplexity reduction for long sequences. We also report the state-of-the-art performance of TAPE in long-context tasks such as passkey retrieval tasks with LLM fine-tuning, and in arithmetic learning.

## 2. Preliminaries

In this work, we aim to design expressive and generalizable positional embeddings for transformers to address complex language tasks. Let $\boldsymbol{X} = \begin{bmatrix} \boldsymbol{x}_1 \cdots \boldsymbol{x}_N \end{bmatrix}^\top \in \mathbb{R}^{N \times C}$ represent the input sequence of tokens, where $N$ is the context length and $C$ is the feature dimension, and $\boldsymbol{M} \in \{0, 1\}^{N \times N}$ denotes the cross-token dependencies, often used as attention mask in transformers. Transformers learn token representations using the attention mechanism (Vaswani et al., 2017),

which propagates information across tokens by computing pairwise correlations. Since pure attention is inherently permutation-equivariant, language models integrate positional information into the attention computation to differentiate tokens based on their positions.

### 2.1. High-Dimensional Features as Positional Encoding

One common approach is to leverage high-dimensional features to represent positions. Positional encoding can be formulated as a series of embeddings attached to each token index $e_1 \cdots e_N$, with the shape of $e_i$ determined by the specified positional encoding schemes. When computing the attention value, the pre-softmax attention value can be in general formulated as below[1]. For every $i, j \in [N]$ such that $\boldsymbol{M}_{i,j} = 1$:

$$\alpha_{i,j} = q(\boldsymbol{x}_i, \boldsymbol{e}_i)^\top k(\boldsymbol{x}_j, \boldsymbol{e}_j), \tag{1}$$

where $q(\cdot, \cdot)$ and $k(\cdot, \cdot)$ are generalized query and key transformations that incorporate positional features. The original transformer paper (Vaswani et al., 2017) assigns each absolute token index a vector of length identical to token embeddings, either learnable or fixed as sinusoidal waves: $\boldsymbol{e}_i \in \mathbb{R}^C$. The query and key transformations directly add the positional information into token features at the first layer: $q(\boldsymbol{x}_i, \boldsymbol{e}_i) = \boldsymbol{W}_Q(\boldsymbol{x}_i + \boldsymbol{e}_i)$ and $k(\boldsymbol{x}_j, \boldsymbol{e}_j) = \boldsymbol{W}_K(\boldsymbol{x}_j + \boldsymbol{e}_j)$ for some query and key matrices $\boldsymbol{W}_Q, \boldsymbol{W}_K \in \mathbb{R}^{C \times C}$. Shaw et al. (2018) introduces learnable embeddings for relative distances, which are applied to the key vector during attention computation. More recently, Rotary Position Encoding (RoPE) (Su et al., 2024) has gained widespread adoption in modern LLMs (Touvron et al., 2023a;b; Biderman et al., 2023; Chowdhery et al., 2023; Jiang et al., 2023). RoPE encodes absolute positions using a series of block-wise rotation matrices $\boldsymbol{E} \in \mathbb{R}^{N \times C/2 \times 2 \times 2}$, while implicitly capturing relative distances during dot-product attention. Formally, the positional embeddings and the transformation $q(\cdot, \cdot)$ are defined as shown below, with $k(\cdot, \cdot)$ adhering to a similar formulation:

$$\boldsymbol{q}(\boldsymbol{x}_i, \boldsymbol{e}_i) = \boldsymbol{R}_i \boldsymbol{W}_Q \boldsymbol{x}_i, \quad \boldsymbol{R}_i = \mathrm{diag}(\boldsymbol{e}_{i,1}, \cdots, \boldsymbol{e}_{i,C/2}),$$

$$\boldsymbol{e}_{i,m} = \begin{bmatrix} \cos(\theta_m i) & -\sin(\theta_m i) \\ \sin(\theta_m i) & \cos(\theta_m i) \end{bmatrix}, \tag{2}$$

where $\mathrm{diag}(\cdot)$ constructs a block-diagonal matrix by concatenating the arguments on the diagonal. In RoPE, the hyper-parameters $\theta_m$ ranges from $\theta_m = -10000^{2m/C}, m \in [C/2]$. Subsequent works explore methods to extend the context length for RoPE-based LLMs through the adoption of damped trigonometric series (Sun et al., 2022), positional interpolation (Chen et al., 2023a) and adjustments to coefficients $\{\theta_m\}_{m \in [C/2]}$ (r/LocalLLaMA, 2023; Peng et al., 2023b; Liu et al., 2023).

---

[1] For simplicity, we ignore the denominator $\sqrt{F}$ by default.

### 2.2. Attention Bias as Positional Encoding

An alternative method for encoding positional information involves applying a bias to the attention map, conditioned on the relative distances between tokens during the attention computation. The pre-softmax attention value with bias can be formulated as:

$$\alpha_{i,j} = (\boldsymbol{W}_Q \boldsymbol{x}_i)^\top (\boldsymbol{W}_K \boldsymbol{x}_j) + b(i,j), \tag{3}$$

where $b(i,j) : \mathbb{N} \times \mathbb{N} \to \mathbb{R}$ is a bias regarding the token indices $i$ and $j$. Many existing positional encoding methods can be interpreted as various instantiations of $b(i,j)$. We follow Li et al. (2023) to summarize a few examples. *(i)* In T5 (Raffel et al., 2020), $b(i,j) = r_{\min\{i-j, L_{max}\}}$, where $L_{max}$ denotes the maximal relative distance considered, and $\{r_i \in \mathbb{R} : i \in [0, L_{max}]\}$ are learnable scalars. *(ii)* Alibi (Press et al., 2021b) simplifies the bias term to $b(i,j) = -r|i - j|$, where $r > 0$ is a hyperparameter that acts as the slope, imposing a linear decay pattern based on the relative distance. *(iii)* Kerple (Chi et al., 2022a) enforces a logarithmic or power decay rate: $b(i,j) = -r_1 \log(1 + r_2|i - j|)$ and $b(i,j) = -r_1|i - j|^{r_2}$ respectively, where $r_1, r_2 > 0$ are hyperparameters. *(iv)* FIRE (Li et al., 2023) learns a neural network with parameters $\boldsymbol{\theta}$ to model the bias: $b(i,j) = f_{\boldsymbol{\theta}}(\psi(i-j)/\psi(\max\{i, L\}))$, where $\psi(x) = \log(cx + 1)$, and $L > 0$ is a hyperparameter.

## 3. Our Approach

### 3.1. Motivations and Design Principles

In the paper, we interpret the attention mechanism as an addressing system, where row-wise attention scores can be viewed as an indicator vector locating important tokens in the context to inform predictions for the current token. The underlying addressing mechanisms include both content-based addressing, which locates tokens via feature similarity, and position-based addressing, which leverages positional encodings to extract location-based information. Content-based addressing is often prioritized in language modeling – which is evidenced by a series of simplifications on positional encoding in the literature (Press et al., 2021b; Haviv et al., 2022; Wang et al., 2024a; Kazemnejad et al., 2024) – due to the fact that natural language semantics primarily depend on the meaning of constituent words rather than their arrangement order (Sinha et al., 2021). However, position-based addressing can sometimes be crucial for many advanced tasks. Ebrahimi et al. (2024) demonstrates that in arithmetic tasks (Lee et al., 2023), a token's position is as important as its value. An ideal attention map for performing addition needs to exclusively rely on token indices.

Moreover, we observe that the interaction between token features and positional embeddings is lacking in current transformers. Golovneva et al. (2024) demonstrate that in-

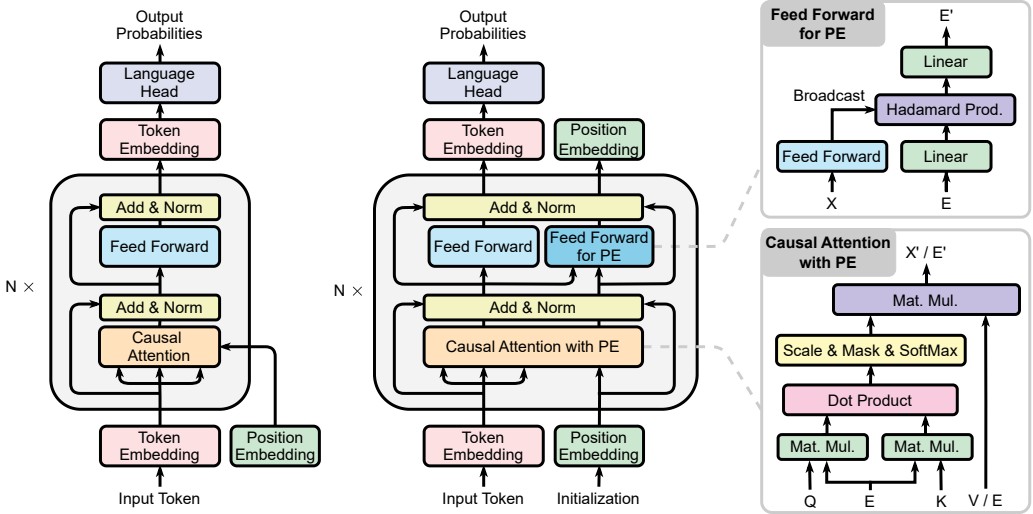

(a) Traditional position embedding.    (b) TAPE with enhanced causal attention and feed forward layers.

*Figure 1.* Overview of our proposed TAPE in standard decoder-only architecture. Different from traditional positional encoding, TAPE represents and updates positional features layer-wisely through interactions and joint training with token representations.

corporating the interplay between context and positional information allows for more flexible addressing, leading to improvements in complex compositional tasks such as algorithm execution and logical reasoning (Liu et al., 2024). In domains with data characterized by intricate geometries, such as graphs and point clouds, capturing the interaction between node or point features and their geometric locations is essential for effectively learning structural patterns relevant to tasks (Wang et al., 2024b; Huang et al., 2024). Algorithmic reasoning requires attention to be endowed with similar properties to internalize representations of computation graphs in order to emulate corresponding algorithms for solving specified tasks (Elhage et al., 2021; Ameisen et al., 2025). However, Merrill & Sabharwal (2023) shows that purely content-based addressing attention is subject to a limited algorithm class.

Based on above observations, we aim to establish a more expressive family of positional encoding, which can be effectively informed by the context to facilitate position-based addressing in LLMs. The main idea is to customize attention and MLP modules in transformers such that they can iteratively update positional embeddings at each layer with sequence content, and use the updated embeddings as the positional encoding for the next layer. We formally outline a couple of key design principles below.

**General Formulation.**    Let a tuple $(\boldsymbol{X}, \boldsymbol{E}, \boldsymbol{M})$ represent a sequence, where $\boldsymbol{X} \in \mathcal{X} \subseteq \mathbb{R}^{N \times C}$ are the token features, $\boldsymbol{E} \in \mathcal{E} \subseteq \mathbb{R}^{N \times D}$ are the positional embeddings, $\boldsymbol{M} \in \mathcal{M} = \{0,1\}^{N \times N}$ encodes the intra-data dependencies, often used as attention mask. For causal language modeling, $\boldsymbol{M}$ becomes constantly a binary lower-triangular matrix ($\boldsymbol{M}_{i,j} \neq 0$ if and only if $j \leq i$). We define a general-

ized transformer block consisting of two separate operations: *token mixing* and *position contextualization*. The token mixing is formulated as a function $f : \mathcal{X} \times \mathcal{E} \times \mathcal{M} \to \mathcal{X}$, which combines token features and positional embeddings to represent each token. The *position contextualization* $g : \mathcal{X} \times \mathcal{E} \times \mathcal{M} \to \mathcal{E}$ encodes the context information into the positional embeddings. A standard transformer layer does not have a component corresponding to $g$ that layer-wisely represents position encodings.

**Tensorial Positional Encoding.**    Our first enhancement extends positional encodings to a multi-dimensional format and diversifies their coupling with token features to allow for richer interactions among token and position representations, drawing inspiration from positional encodings used in geometric learning (Deng et al., 2021; Wang et al., 2024b; Huang et al., 2024). We first divide the hidden dimension into $M$ blocks, resulting $\boldsymbol{X} = [\boldsymbol{x}_1, \cdots, \boldsymbol{x}_N]^\top \in \mathbb{R}^{N \times M \times B}$ with $\boldsymbol{x}_i \in \mathbb{R}^{M \times B}$ and $B = C/M$. We propose to assign each block $L$-many $R$-dimension positional embeddings. Therefore, we reorganize positional embeddings as $\boldsymbol{E} = [\boldsymbol{e}_1, \cdots, \boldsymbol{e}_N]^\top \in \mathbb{R}^{N \times M \times L \times R}$ with $\boldsymbol{e}_i \in \mathbb{R}^{M \times L \times R}$ and $D = M \times L \times R$.

**Equivariance Principles.**    Second, we establish two fundamental criteria for the design of functions $f$ and $g$. Conceptually, by representing each token as a tuple comprising its token and positional embedding, the entire sequence can be viewed as an unordered set. This implies that permuting these tuples arbitrarily will not alter the outputs of $f$ and $g$, aside from a corresponding change in order (Zaheer et al., 2017; Lee et al., 2019). We note that this is an intrinsic property of standard attention. Furthermore, we aim for the positional embeddings to effectively model relative dis-

tances, necessitating that $f$ remains invariant to translations in the token positions (Sun et al., 2022). This invariance can be achieved by structuring $f$ and $g$ to depend on the positional embeddings in a manner invariant and equivariant, respectively, to orthogonal transformations along the last dimension (Villar et al., 2021). Formally, let us denote $\Pi(N)$ as a permutation group over $N$ elements, and $O(R)$ as an orthogonal group over the $R$-dimension Euclidean space. The two aforementioned criteria require $f$ and $g$ to satisfy that: for $\forall \boldsymbol{P} \in \Pi(N), \boldsymbol{R} \in O(R)$,

$$f(\boldsymbol{PX}, \boldsymbol{PER}, \boldsymbol{PMP}^\top) = \boldsymbol{P}f(\boldsymbol{X}, \boldsymbol{E}, \boldsymbol{M}) \quad (4)$$

$$g(\boldsymbol{PX}, \boldsymbol{PER}, \boldsymbol{PMP}^\top) = \boldsymbol{P}g(\boldsymbol{X}, \boldsymbol{E}, \boldsymbol{M})\boldsymbol{R} \quad (5)$$

where left-multiplication of $\boldsymbol{P} \in \Pi(N)$ permutes on the first dimension of $\boldsymbol{X}$ and $\boldsymbol{E}$, while right-multiplication of $\boldsymbol{R} \in O(R)$ applies to the last dimension of tensor $\boldsymbol{E}$. When permutation is applied for the attention mask, it permutes both columns and rows accordingly. We note that attention with RoPE inherently satisfies Eq. 4, with the invariant orthogonal group being $O(2)$. We formalize the merits of position contextualization and group invariance in Sec. 4. In summary, we prove that contextualized positional encoding improves flexibility and adaptability to represent a broader class of algorithms than either raw attention or RoPE. Additionally, orthogonal group equivariance ensures that the positional encodings remain relative, facilitating generalization across varying context lengths.

### 3.2. Contextualized Positional Encoding with Equivariance

In this section, we instantiate design principles discussed in Sec. 3.1 as a practical neural architecture. We note that although there are lots of ways to achieve conditions in Eq. 4 and 5 (Dym & Maron, 2020; Bogatskiy et al., 2020; Yarotsky, 2022), the proposed method focuses on enhancing existing components used in standard transformers with consideration of computational efficiency. We term our proposed approach of informing positional encoding with context through enforcing equivariance as Con**T**extualized Equiv**A**riant **P**ositional **E**ncoding (TAPE).

**Model Structure and Initialization.** We adhere to the conventional architecture of the standard transformer (Vaswani et al., 2017), wherein each layer comprises an attention module for token mixing and a Multi-Layer Perceptron (MLP) for channel mixing. Both the attention and MLP components are tailored to update positional embeddings at each layer. We depict the overall architecture in Fig. 1. The initial positional features may encompass a variety of representations, including but not limited to learnable features (Vaswani et al., 2017), sinusoidal series (Vaswani et al., 2017; Su et al., 2024; Sun et al., 2022), or random Fourier features (Rahimi & Recht, 2007; Yu et al., 2016).

Among these, we select the widely-used sinusoidal series embedding, RoPE (Su et al., 2024), as our initialization, as detailed in Sec. 3.3.

$O(R)$**-Invariant Token Mixing.** In each transformer block, $f$ updates token features through attention and MLP following the principles of permutation-equivariance and $O(R)$-invariance. We define pre-softmax attention value between the $i$-th and $j$-th tokens ($\boldsymbol{M}_{i,j} \neq 0$) as:

$$\begin{aligned} \alpha_{i,j} &= \sum_{m=1}^{M} \alpha_{i,j,m}, \\ \alpha_{i,j,m} &= (\boldsymbol{W}_Q \boldsymbol{x}_i)_m^\top \phi(\boldsymbol{e}_{i,m} \boldsymbol{e}_{j,m}^\top)(\boldsymbol{W}_K \boldsymbol{x}_j)_m, \end{aligned} \quad (6)$$

where $\phi(\cdot) : \mathbb{R}^{L \times L} \to \mathbb{R}^{B \times B}$ can be any function. Permutation-equivariance is inherently preserved in pairwise attention, regardless of the method used to derive attention values. $O(R)$-invariance is achieved by computing the inner product of positional embeddings (Villar et al., 2021; Wang et al., 2022a; 2024b). We note that $O(R)$-invariance stems from the separation of the inner product calculations between features and positional embeddings, in contrast to Vaswani et al. (2017). In practice, we can let $B = L$ and $\phi$ be an identity mapping, which simplifies Eq. 6 to a series of tensor multiplications. After applying attention, a standard MLP layer is employed to transform token embeddings without using positional embeddings.

$O(R)$**-Equivariant Position Contextualization.** The primary contribution of this work is the introduction of a method to condition positional embeddings on sequence content. We employ an $O(R)$-equivariant function $g$ to ensure structure conservation of this update. A key insight is that linearly combining positional coordinates preserves $O(R)$-equivariance, provided the weights are invariant to the orthogonal group (Villar et al., 2021; Wang et al., 2022a; Huang et al., 2024). This observation leads us to leverage attention maps, which capture content-based token relationships, to integrate positional embeddings. Hence, the attention layer can update positional embedding via:

$$\widetilde{\boldsymbol{e}}_{i,m} = \sum_{j=1}^{N} \frac{\boldsymbol{M}_{i,j} \exp(\alpha_{i,j,m})}{\sum_{j'=1}^{N} \boldsymbol{M}_{i,j} \exp(\alpha_{i,j',m})} \boldsymbol{e}_{j,m}, \quad (7)$$

where $\{\tilde{\boldsymbol{e}}_{i,m}\}_{j \in [N], m \in [M]}$ denotes an intermediate output of the attention layer. In practice, we share the attention map between Eq. 6 and 7. We can re-use $\alpha_{i,j,m}$ computed in Eq. 6 because we have shown that attention weights $\alpha_{i,j,m}$ are $O(R)$-invariant.

We further propose a layer similar to the function of MLP, which directly transform matrix-form positional embeddings with token features incorporated. Specifically, we first flatten the first two dimensions of $\widetilde{\boldsymbol{e}}_i \in \mathbb{R}^{M \times L \times R}$ to the

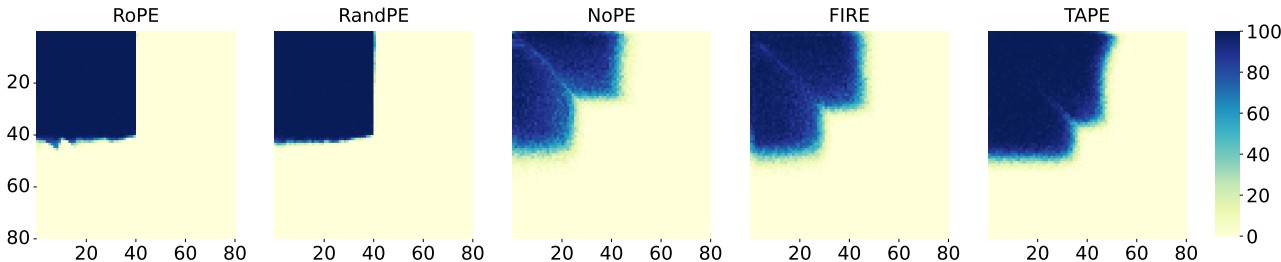

*Figure 2.* Accuracy on addition task between different methods on $2\times$ context length. The x- and y-axes represent the sequence lengths of the two operands respectively. Models are trained on sequence with length up to 40 while tested on sequence with length up to 80. The average accuracy across the heatmap is 26.32%, 26.56%, 22.45%, 26.98% and 32.82% respectively for RoPE (Su et al., 2024), RandPE (Ruoss et al., 2023), NoPE (Kazemnejad et al., 2024), FIRE (Li et al., 2023), and our TAPE.

shape $\mathbb{R}^{ML \times R}$, then apply linear transformation constructed by token features, and finally unflatten the first dimension of the resultant matrix to $\widehat{e}_i \in \mathbb{R}^{M \times L \times R}, \forall j \in [N]$:

$$\widehat{e}_i = \text{unflatten}\left(W_2 \psi(\widetilde{x}_i) W_1^\top \text{flatten}(\widetilde{e}_i)\right), \quad (8)$$

where we denote $\widetilde{x}_i \in \mathbb{R}^C$ as the output of attention after token mixing. Let $I$ be the intermediate dimension, $W_1$ and $W_2$ have shape $ML \times I$. We further define $\psi : \mathbb{R}^C \to \mathbb{R}^{I \times I}$ as a mapping between token features to linear transformations. To reduce computation overhead, we adopt an MLP to transform $\tilde{x}_i$ into an $I$-dimensional vector and form a diagonal matrix with the resultant vector as its diagonal. The detailed computational flow is illustrated in Fig. 4 in Appendix C. By applying linear transformations only to the first two dimensions of $\widetilde{e}_i$, this layer maintains $O(R)$-equivariance.

We summarize the overall geometric properties of our architecture in Proposition 3.1 below.

**Proposition 3.1.** *The proposed TAPE including: (i) attention in Eq. 6 with normal MLP for token mixing, and (ii) attention in Eq. 7 with MLP defined in Eq. 8 for position contextualization, satisfies Eq. 4 and Eq. 5.*

### 3.3. Parameter-Efficient Fine-tuning with TAPE

In this section, we demonstrate that our TAPE can be seamlessly integrated into pre-trained models, enabling Parameter-Efficient Fine-Tuning (PEFT) to enhance position-based addressing in existing architectures. Notably, the widely adopted RoPE (Su et al., 2024) can be considered a special case of TAPE. This can be seen by letting $L = R = 2$ and $e_{i,m,1} = \begin{bmatrix} \cos(\theta_m i) & -\sin(\theta_m i) \end{bmatrix}^\top, e_{i,m,2} = \begin{bmatrix} \sin(\theta_m i) & \cos(\theta_m i) \end{bmatrix}^\top$. With this configuration, Eq. 6 becomes equivalent to Eq. 2. As a result, RoPE can serve as the initialization for TAPE, while the model is further enhanced by incorporating the contextualization component specified in Eq. 7 and 8. This initialization is applied across all experiments, encompassing both pre-training and fine-tuning stages. Specifically, during fine-tuning, to ensure

that the augmented model remains identical to the original at initialization, we follow Hu et al. (2022) by setting the initialization of $W_2$ in Eq. 8 to zero such that all updates to the positional encoding inside the block will then be reset via a residual connection. To allow for parameter-efficient fine-tuning, we enable gradients to update the weights for position encoding contextualization including $W_1, W_2$ and the weights for the post-attention linear layer, while keeping all other parameters frozen.

## 4. Theoretical Analysis

In this section, we formally state the theoretical benefits of our proposed architecture, analyzing two key aspects: (1) its enhanced expressiveness for algorithmic reasoning, enabled by the contextualization mechanism, and (2) its guaranteed stability under equivariance constraints.

**Benefits of Contextualization.** To characterize the expressive power of transformer architectures on reasoning problems, we adopt a circuit-complexity lens following Merrill (2019); Merrill & Sabharwal (2023); Liu et al. (2022); Yang et al. (2025), identifying families of algorithms these models can and cannot simulate. Our premise is that formal reasoning entails multi-step algorithmic computation. Effective reasoning with LLMs therefore requires sufficient capacity to represent these algorithms within the model.

Circuit complexity quantifies the difficulty of different problems, with NC, AC and TC being the key classes. The classes NC, AC, and TC consist of problems (with input size $N$) solvable by circuits of depth $O((\log N)^k)$. Specifically, $\text{NC}^k$ allows only bounded fan-in AND, OR, and NOT gates; $\text{AC}^k$ extends $\text{NC}^k$ by permitting unbounded fan-in gates; and $\text{TC}^k$ further includes unbounded fan-in threshold gates. We refer interested readers to formal definitions in Li et al. (2025); Vollmer (1999); Arora & Barak (2009). It is known that $\text{AC}^0 \subsetneq \text{TC}^0 \subseteq \text{NC}^1$ but it remains open whether $\text{TC}^0 = \text{NC}^1$ is achievable (Cook, 1985).

Below we show the algorithmic expressivity of TAPE

reaches $\mathsf{NC}^1$-complete, which implies all the problems as hard as a $\mathsf{NC}^1$-complete problem or within the $\mathsf{NC}^1$ class can be solved by transformers with TAPE.

**Theorem 4.1** (Circuit Representational Power). *There exists a constant-depth and constant-width transformer equipped with TAPE and satisfying Eqs. 4 and 5 such that it is capable of solving all $\mathsf{NC}^1$-complete problem under $\mathsf{AC}^0$-reductions.*

It is known that the representation capacity of constant-layer transformers is upper bounded by $\mathsf{TC}^0$. Theorem 4.1 shows a clear separation in expressivity between transformers with and without TAPE unless $\mathsf{TC}^0 = \mathsf{NC}^1$. Equipping attention with TAPE can potentially enhance its computational power.

The proof of Theorem 4.1 shows TAPE can solve $\mathsf{NC}^1$-complete state-tracking problem. When understanding LLM reasoning from the automaton perspective (Merrill, 2019), state-tracking ability is essential for modeling sequential computations and carrying out formal logic entailment (Merrill et al., 2024; Peng et al., 2025; Grazzi et al., 2024). Our proof also highlights the importance and the role of layer-wise update of positional encoding, contextualized positional-informed token mixing. The multi-layer construction illustrates how layer-wise update of positional encoding progresses positional and tokens features into the positional encoding, which learns to be a more complex representation encoding inherent algorithmic reasoning structures.

This result is largely inspired by the concurrent work of Yang et al. (2025). It proves a seminal result showing that positional encodings constructed by accumulating input-conditioned Householder matrices (Yang et al., 2024) can uniformly represent all algorithms within $\mathsf{NC}^1$ class. Our proof connects the accumulation of Householder matrices with our proposed geometric constraints.

**Importance of Equivariance.** It has been shown that positional embeddings that encode relative distances among tokens are crucial for ensuring long-context capabilities in LLMs (Li et al., 2023). Although initializing positional encodings with RoPE or random Fourier features at the first layer guarantees relativity under inner product (see Eq. 6), it remains unclear whether this property is preserved after the embeddings are updated in intermediate layers. To address this concern, we provide the following justification.

**Proposition 4.2** (Relativity of position encoding). *Suppose a transformer consists of $f, g$ satisfying Eqs. 4 and 5. Given inputs $(\boldsymbol{X}, \boldsymbol{E})$ with position encoding $\boldsymbol{E}$ initialized by RoPE or random Fourier features, then the transformer is invariant to shift on token indices.*

Proposition 4.2 highlights the importance of the equivariance principle with respect to the orthogonal group. Under $O(R)$-equivariance, even after positional encodings are updated with token features, their effect on attention is still determined by the relative distances between token positions (Sinha et al., 2022). This property also ensures stability and supports generalization beyond the training context length.

# 5. Experiments

In this section, we first validate our method on arithmetic tasks, which relies on better position-addressing ability for prediction (Sec. 5.1). We also show our effectiveness in natural languages, in both pre-training (Sec. 5.2) and fine-tuning case (Sec. 3.3). More experiments, visualization, and model interpretation can be found in Appendix D and E.

## 5.1. Arithmetic Learning

As demonstrated by prior research (Lee et al., 2023; Zhou et al., 2024), even large transformer models struggle with arithmetic tasks. Recent studies suggest that this limitation may stem from their constrained position-addressing capabilities (Ebrahimi et al., 2024). In particular, arithmetic tasks treat every digit as equally important to the equation, regardless of its distance from the output. In contrast, traditional positional embeddings for language tasks often assume a distance-decay effect, where words farther apart have less significance in the output. Positional contextualization potentially addresses this by dynamically reweighting positional importance based on the task context. To evaluate the ability of LLMs of performing arithmetic tasks with our position embedding, we use the Addition Bucket 40 dataset (McLeish et al., 2024a) which contains 20 million samples with $i \times i$ ($i < 40$) operand lengths. We train transformers from scratch using the arithmetic data, and during evaluation, we sample 100 samples for each pair of operand lengths. Following the existing attempt (McLeish et al., 2024a), the operands in the training set are not necessary to have the same length, but the maximum length of two operands are the same. We then report model accuracy for each $(i, j)$ length pair. Note that accuracy is measured strictly, counting only exact matches of all output digits as correct. The transformers are standard decoder-only architecture with config detailed in Appendix B. We compare our method with four baselines, including RoPE (Kitaev et al., 2020), RandPE (Ruoss et al., 2023) NoPE (Kazemnejad et al., 2024), and FIRE (Li et al., 2023).

The heatmaps further demonstrate TAPE's superior generalization to longer sequences, as indicated by the concentrated dark-colored regions representing higher accuracy across a wider range of operand lengths. TAPE outperforms other methods with the highest average accuracy of 32.82%. Compared to FIRE, which achieves 26.98% and previously held the strongest length generalization in arithmetic tasks (McLeish et al., 2024a; Zhou et al., 2024), TAPE shows a remarkable 21.6% relative improvement. This shows TAPE's effectiveness in maintaining accuracy

*Table 1.* Performance comparison on seven datasets from SCROLLS benchmark. For all tasks, the performance is better if the reported metric is higher (↑). We highlight the top-performing methods via **bold** font.

|  | QAS | CNLI | NQA | QuAL | QMS | SumS | GovR |
|---|---|---|---|---|---|---|---|
| Metric (%) | F1 (↑) | EM (↑) | F1 (↑) | EM (↑) | Rgm (↑) | Rgm (↑) | Rgm (↑) |
| Median length | 5472 | 2148 | 57829 | 7171 | 14197 | 9046 | 8841 |
| RoPE | 8.39 | 65.00 | 1.77 | 0.04 | 6.34 | 5.63 | 9.71 |
| ALiBi | 8.25 | 69.62 | 4.11 | 0.0 | 9.92 | 9.78 | **18.81** |
| RandPE | **13.44** | 62.01 | 4.63 | 0.38 | 8.43 | 8.31 | 8.93 |
| FIRE | 3.41 | 71.26 | 0.48 | 1.25 | 8.78 | 7.42 | 11.03 |
| xPos | 9.02 | 71.75 | 4.83 | 0.24 | 10.73 | 9.38 | 16.38 |
| TAPE (Ours) | 11.52 | **72.80** | **6.79** | **11.60** | **12.42** | **10.34** | 15.18 |

as sequence lengths increase, making it particularly suitable for long-range dependency tasks.

## 5.2. Pre-Training from Scratch

Pre-training a language model on a corpus followed by fine-tuning on downstream tasks is the standard methodology for evaluating the performance of positional embeddings in prior studies (Li et al., 2023; He et al., 2024). Similarly, we first pre-train transformers with 1024 context window from scratch, using C4 dataset (Raffel et al., 2020), and then fine-tune those models in long-context benchmark SCROLLS (Shaham et al., 2022). We report three evaluation metrics for seven different tasks: unigram overlap (F1) for Qasper and NarrativeQA, and exact match (EM) for QuALITY (QAS) and ContractNLI (CNLI), and Rgm score (the geometric mean of ROUGE-1,2,L) for the three summarization tasks: GovReport (GovR), QMSum (QMS), and SummScreenFD (SumS). We compare our methods with RoPE (Kitaev et al., 2020), ALiBi (Press et al., 2021a), RandPE (Ruoss et al., 2023), FIRE (Li et al., 2023) and xPos (Sun et al., 2022), and report the results in Tab. 1.

Our method consistently outperforms all baselines, demonstrating significant improvements, particularly in scenarios with longer context lengths, as observed in QuAL and NQA. In terms of overall performance, xPos is the closest competitor to TAPE. While FIRE, RandPE, and ALiBi exhibit good results on a few datasets, they fall short across the board. RoPE struggles with all long-context datasets.

## 5.3. Context Window Extension by PEFT

We extend the context window of the pre-trained Llama2 7B model (GenAI, 2023) from 4096 to 8192, using the Redpajama (Computer, 2023). For validation, we then compare the perplexity on sequence of length 8192, on the cleaned ArXiv Math proof-pile dataset (Azerbayev et al., 2022; Chen et al., 2023a) and the book corpus dataset PG19 (Rae et al., 2019). To further evaluate the models' performance of long context understanding, we report the accuracy of fine-tuned

*Table 2.* Evaluation on perplexity across 1k to 8k context lengths. Lower perplexity means better performance (↓). Top results are marked **bold**. Each model is first pre-trained from scratch and later fine-tuned on the downstream long-context datasets.

| Method | 1024 | 2048 | 4096 | 8192 |
|---|---|---|---|---|
| **Proof-pile** | | | | |
| LoRA | 3.828 | 3.369 | 3.064 | 2.867 |
| LongLoRA | 3.918 | 3.455 | 3.153 | 2.956 |
| Theta Scaling | 3.864 | 3.415 | 3.121 | 2.934 |
| TAPE (Ours) | **3.641** | **3.196** | **2.901** | **2.708** |
| **PG-19** | | | | |
| LoRA | 9.791 | 9.098 | 8.572 | 8.199 |
| LongLoRA | 9.989 | 9.376 | 8.948 | 8.645 |
| Theta Scaling | 9.257 | 8.640 | 8.241 | 7.999 |
| TAPE (Ours) | **8.226** | **7.642** | **7.278** | **7.063** |

models on passkey retrieval task which has been adopted by many literature (Chen et al., 2023b;a; Tworkowski et al., 2024). We choose a popular open-sourced LLM Llama2 7B (Touvron et al., 2023b), which uses RoPE, as the base model and extend it to the 8192 context length. Three baselines are selected to compare to our TAPE method: vanilla LoRA (Hu et al., 2022), LongLoRA (Chen et al., 2023b), Theta Scaling (Liu et al., 2023).

As shown in Tab. 2, TAPE consistently outperforms the other methods across all context lengths on both the Proof-pile and PG19 datasets. On Proof-pile, TAPE achieves a perplexity of 3.641 at 1024 tokens, improving over LoRA (3.828), LongLoRA (3.918), and Theta Scaling (3.864). At 8192 tokens, TAPE's advantage grows, reaching 2.708, surpassing LongLoRA (2.956), LoRA (2.867), and Theta Scaling (2.934). Similarly, on PG19, TAPE achieves 8.226 at 1024 tokens, improving up to 18.3% over competitors. At 8192 tokens, TAPE reaches 7.063, further showing superiority, especially at longer context lengths.

We also evaluate the passkey retrieval accuracy of our model, following Landmark Attention (Mohtashami & Jaggi, 2023), which has also been adopted by other literature (Chen et al., 2023a; Tworkowski et al., 2024; Chen et al., 2023b). In

*Figure 3.* Accuracy on passkey retrieval from 1k to 8k context length with Llama2 7B. We adopt the parameter-efficient fine-tuning strategy for TAPE (see Sec. 3.3). In contrast to other parameter-efficient fine-fining methods (e.g. LoRA (Hu et al., 2022) and LongLoRA (Chen et al., 2023b)), TAPE achieves no accuracy drop at 8k context length. TAPE performs even on par with full-parameter tuning with Theta Scaling (Liu et al., 2023).

*Table 3.* Comparison of FLOPS, MACs, and the number of parameters for models with different position embeddings.

| Method | TAPE | RoPE | FIRE | T5's relative bias |
|---|---|---|---|---|
| FLOPS (G) | 365.65 | 321.10 | 331.97 | 321.10 |
| MACs (G) | 180.69 | 160.46 | 165.69 | 160.46 |
| Params. (M) | 155.33 | 154.89 | 154.90 | 154.90 |

*Table 4.* System measurement. We report execution time per step in the **Time** row and iteration per second in the **Throughput** row. The values are averaged over 100 inference steps.

| Method | TAPE | | RoPE | FIRE | T5's relative bias |
|---|---|---|---|---|---|
| | w/ Fusion | w/o Fusion | | | |
| Time ($\times 10^{-4}$) | 2.56 | 5.63 | 2.08 | 5.56 | 6.90 |
| Throughput | 3910 | 1775 | 4810 | 1799 | 1449 |
| Flash Attention | ✓ | ✓ | ✓ | ✗ | ✗ |

this task, the models are required to locate and retrieve a random passkey hidden in a long document. We test the passkey retrieval accuracy ranging from 1k to 8k. The results of long-context passkey retrieval task is presented in Fig. 3. As shown, TAPE consistently achieves near-perfect accuracy across all context lengths, outperforming other methods. Theta Scaling shows a relatively stable performance while LoRA and LongLoRA exhibit fluctuating and lower accuracy. Notably, Theta Scaling is widely employed in popular open-source long-context models like Llama3 8B Instruct 262k (AI@Meta, 2024) and MistralLite (AWS, 2024). TAPE demonstrates a similar superior capability to be applied in long-context tasks.

### 5.4. Efficiency Analysis

In this subsection, we analyze the complexity of our methods in comparison to traditional position embedding techniques. Using the models from the pretraining experiment in Sec. 5.2, we report three key metrics: FLOPs, MACs, and the number of parameters. The metrics are evaluated with a batch size of 1 and sequence length 1024. As shown in Tab. 3, our architectural modifications introduce a negligible increase in FLOPs, MACs and number of parameters, compared to the standard Transformer with RoPE. Moreover, our TAPE is fully compatible with Flash Attention (Dao et al., 2022; Dao, 2024a), a widely adopted accelerated attention mechanism with IO-awareness, which introduces extra efficiency.

For simplicity, we evaluate the running time of attention layers with different position embedding methods on a sin-

gle A100 GPU. We run 100 inference steps and report the average execution time. Both RoPE and TAPE leverage the acceleration provided by Flash Attention (Dao, 2024b), whereas FIRE and T5's relative bias are not fully compatible with Flash Attention, as it currently lacks support for gradient computation in relative bias. In contrast, we observe that the computations for position embeddings and token features in TAPE are highly parallelizable, making it suitable for further acceleration using kernel fusion techniques. To capitalize on this, we implemented a version of TAPE with kernel fusion, referred to as TAPE w/ Fusion. As shown in Tab. 4, the efficiency of the original TAPE (w/o Fusion) already surpasses T5's relative bias and is comparable to FIRE. With additional kernel fusion applied, TAPE achieves a 2.2× speedup, approaching the efficiency of RoPE with Flash Attention.

## 6. Conclusion

This paper introduced TAPE, a framework that enhances transformer models by contextualizing positional embeddings with sequence content across layers. Through incorporating permutation and orthogonal equivariance, we ensured stability and adaptability in positional encoding updates. TAPE can also be easily integrated into existing models, introducing negligible computation and inference overhead. Extensive experiments confirmed TAPE's effectiveness in both arithmetic reasoning and long context language modeling tasks. One current limitation lies in our exclusive focus on decoder-only models, with limited training scale.

## Acknowledgements

The work reported in this paper was substantially performed using the Princeton Research Computing resources at Princeton University, which is a consortium of groups led by the Princeton Institute for Computational Science and Engineering (PICSciE) and Research Computing. PW thanks Kaiyue Wen and Songlin Yang for discussions on the role of positional encoding in circuit representational power, Shenghao Yang for mentioning the relevance with algorithmic reasoning, and Chuanyang Zheng for discussions on context-aware position encodings. PW and PL also thank Yinan Huang and Siqi Miao for discussing positional encoding for geometric data. PI Li is supported by NSF awards IIS-2239565 and IIS-2428777 for this project. PI Wang is in part supported by NSF Awards 2145346 (CAREER), 02133861 (DMS), and the NSF AI Institute for Foundations of Machine Learning (IFML).

## Impact Statement

This paper presents work whose goal is to advance the field of Machine Learning. There are many potential societal consequences of our work, none of which we feel must be specifically highlighted here.

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

# A. More Related Work

**Context Length Extrapolation.** The length extrapolation ability of transformers are limited mainly in two aspects: (1) the high memory usage caused by quadratic memory usage; and (2) the poor generalizability to unseen sequence length during inference. To address the memory usage during long sequences training, LongLoRA (Chen et al., 2023b) introduced shifted sparse attention and leveraged parameter-efficient tuning. LoCoCo (Cai et al., 2024) introduce a KV cache compression mechanism. To help generalizability of positional embedding to unseen sequence length, Chen et al. (2023a) explores zero-shot linear interpolation on rotary embedding; r/LocalLLaMA (2023); Peng et al. (2023b) enhance simple interpolation by retaining high-frequency encoding ability; Liu et al. (2023) investigates the relationship between rotary base and extrapolation ability. While the previously mentioned methods focus primarily on extending rotary positional embeddings, Li et al. (2023) introduced a functional relative position encoding framework that enhances generalization to longer contexts. However, these methods generally impose a fixed pattern on attention maps, often adopting a decaying pattern based on distance. In contrast, we propose a learnable and generic position encoding framework that primarily focuses on arithmetic reasoning.

**Equivariant Learning.** Equivariant machine learning is a broad field that leverages task-specific symmetries to introduce inductive biases into neural networks, reducing learning complexity and improving generalization. Here, we focus on foundational works in this domain that are highly relevant and provide key motivation for our study. Prior research has primarily focused on data representations with intrinsic symmetries, such as graphs (Satorras et al., 2021; Schütt et al., 2021; Batzner et al., 2022; Maron et al., 2018), hyper-graphs (Wang et al., 2022b; Chien et al., 2021), and point clouds (Zaheer et al., 2017; Fuchs et al., 2020; Thomas et al., 2018; Hoogeboom et al., 2022), which primarily require permutation equivariance. Recent progress models graph representation learning as a joint invariance of permutation and orthogonal groups (Wang et al., 2022a; Huang et al., 2023; Wang et al., 2024b). Beyond geometric data, another stream of work (Worrall et al., 2017; Zhang, 2019; Weiler & Cesa, 2019; Cohen et al., 2019) ensures symmetric inputs yield consistent outputs, reflecting the same label under symmetric transformations. For instance, Worrall et al. (2017) introduce rotation-equivariant feature transformations after mapping images to the continuous fourier domain, while Zhang (2019) enhance translation invariance in CNNs by incorporating a low-pass filter in the pooling layer. To the best of our knowledge, we are the first to introduce equivariance in language models, bridging the geometric properties of positional embeddings to modeling complex language tasks (e.g., retrieval and reasoning) and learning stability.

**Concurrent works on Contextualized Positional Encoding.** Recent studies have moved beyond fixed absolute or relative encodings toward context-adaptive position signals. DAPE (Zheng et al., 2024) pioneers the design of context-dependent positional embeddings, where positional information is decomposed into a learned prior modulated by the current attention context. This yields a data-adaptive signal that improves both in-length performance and length extrapolation. A subsequent work CoPE (Golovneva et al., 2024) proposes a positional contextualization that accumulates intervals determined by token features to represent token positions in the sequence. PaTH Attention (Yang et al., 2025), parameterizes positional effects as an accumulated product of input-conditioned Householder-like transformations, offering a compact parallel formulation and a FlashAttention-style blockwise implementation. Importantly, Yang et al. (2025) highlights the role of positional encoding in enhancing the algorithmic representational power of transformers and strengthening their computational completeness. Our theoretical results are highly motivated by Yang et al. (2025).

**Algorithmic Reasoning and Geometric Learning.** A large volume of work studies training neural networks to execute algorithms. In these settings, models take as input various data structures – often encoded as graphs – and compute solutions to tasks such as path finding, graph connectivity, and dynamic programming (Veličković & Blundell, 2021; Ibarz et al., 2022; Yan et al., 2020; Veličković et al., 2022; Dudzik & Veličković, 2022; Bevilacqua et al., 2023; Georgiev et al., 2023; Engelmayer et al., 2024; Diao & Loynd, 2022; Tang et al., 2020). A recent line of work investigates transformers' capabilities as general-purpose algorithmic performers (Delétang et al., 2022; Butoi et al., 2024). More recently, Positional Attention (de Luca et al., 2025) formalizes transformer architectures whose attention weights depend solely on positional encodings rather than token features. At a high level, attention endows models with the capacity to realize computation graphs over tokens (Elhage et al., 2021; Ameisen et al., 2025). The structure of our proposed TAPE can be viewed as a natural way of embedding the discrete, symbolic, and input-dependent graph structures of algorithms into continuous representations with orthogonal-group equivariance. A similar transformation has also been seen in designing graph transformers with Laplacian embeddings Dwivedi & Bresson (2020); Wang et al. (2022a). The duality between graph topological representation and geometric embeddings has been investigated in Wang et al. (2024b).

# B. Deferred Proofs

**Notations.** The vector $\boldsymbol{\delta}_i$ denotes the $i$-th canonical basis vector, i.e., a vector with $1$ in the $i$-th position and $0$ elsewhere. Its length can be often inferred by context without cause of ambiguity. We denote the permutation group over $N$ entries as $\Pi(N)$, and orthogonal group of $R$ dimension as $O(R)$. Associated with a permutation matrix $\boldsymbol{P} \in \Pi(N)$, we represent the *permutation map* as a function $\pi : [N] \to [N]$, i.e., $\boldsymbol{P}_i = \boldsymbol{\delta}_{\pi(i)}$. We also work with the swap (i.e., transposition) group $S(N) \subset \Pi(N)$ which exchanges up to two elements in $[N]$. Note that $S(N)$ is a generating set of $\Pi(N)$.

## B.1. Formal Statement and Proof for Theorem 4.1

In this section, we formalize the statement of Theorem 4.1 with the hypothesis class yielded by transformers with TAPE and the precise definition of representational power of circuits.

### B.1.1. ARCHITECTURAL FAMILY

First of all, we define a family of transformer architectures equipped with generalized TAPE subject to invariance principles (Eqs. 4 and 5). Recap that we denote a sequence of token features as $\boldsymbol{X} \in \mathcal{X} \subseteq \mathbb{R}^{N \times C}$, positional encoding as $\boldsymbol{E} \in \mathcal{E} \subseteq \mathbb{R}^{N \times ML \times R}$, and masking matrix $\boldsymbol{M} \in \mathcal{G} \subseteq \mathbb{R}^{N \times N}$. Without loss of generality, we assume the shape of token and positional representations is unchanged in hidden layers. Following our general formulation in Sec. 3.1, each layer of such a transformer consists of two components: *(1)* token mixing layer $f^{(l)} : (\boldsymbol{X}^{(l-1)}, \boldsymbol{E}^{(l-1)}, \boldsymbol{M}) \mapsto \boldsymbol{X}^{(l)}$, and *(2)* position contextualization layer $g^{(l)} : (\boldsymbol{X}^{(l-1)}, \boldsymbol{E}^{(l-1)}, \boldsymbol{M}) \mapsto \boldsymbol{E}^{(l)}$, where we use superscript to denote the layer index. The token mixing layer follows a standard design of the attention block with residual connections and an element-wise feedforward mapping: $f^{(l)} = f_{\mathsf{FFN}}^{(l)} \circ f_{\mathsf{Attn}}^{(l)}$ such that $f_{\mathsf{Attn}}^{(l)}(\boldsymbol{X}, \boldsymbol{E}, \boldsymbol{M})_i = \sum_{j=1}^{N} A^{(l)}(\boldsymbol{X}, \boldsymbol{E}, \boldsymbol{M})_{ij} \boldsymbol{W}_V^{(l)} \boldsymbol{x}_j + \mathsf{Res}_{f_{\mathsf{Attn}}}^{(l)} \boldsymbol{x}_i$ where $\mathsf{Res}_*^{(l)} \in \{0, 1\}$ denotes whether skip connection is adopted across the module $*$ at the $l$-th layer, and $A^{(l)}$ is a conventional attention modulated by positional encoding:

$$A^{(l)}(\boldsymbol{X}, \boldsymbol{E}, \boldsymbol{M})_{ij} = \frac{\boldsymbol{M}_{ij} \exp\left(\boldsymbol{x}_i^\top \boldsymbol{W}_Q^{(l)\top} \phi^{(l)}(\boldsymbol{e}_i, \boldsymbol{e}_j) \boldsymbol{W}_K^{(l)} \boldsymbol{x}_j\right) \boldsymbol{W}_V \boldsymbol{x}_j}{\sum_{j'=1}^{N} \boldsymbol{M}_{ij'} \exp\left(\boldsymbol{x}_i^\top \boldsymbol{W}_Q^{(l)\top} \phi^{(l)}(\boldsymbol{e}_i, \boldsymbol{e}_{j'}) \boldsymbol{W}_K^{(l)} \boldsymbol{x}_{j'}\right)}.$$

We denote $\boldsymbol{W}_K^{(l)}, \boldsymbol{W}_Q^{(l)}, \boldsymbol{W}_V^{(l)} \in \mathbb{R}^{C \times C}$ as key, query, and value matrices, and $\phi^{(l)} : \mathbb{R}^{M \times L \times R} \times \mathbb{R}^{M \times L \times R} \to \mathbb{R}^{C \times C}$ injects information of positional encoding into attention logits. In addition, $f_{\mathsf{FFN}}^{(l)}(\boldsymbol{X})_i = \mathsf{FFN}^{(l)}(\boldsymbol{x}_i) + \mathsf{Res}_{f_{\mathsf{FFN}}}^{(l)} \boldsymbol{x}_i$ with $\mathsf{FFN}^{(l)}$ as an element-wise MLP of two layers, independent of positional encoding. Clearly, permutation equivariance holds due to the nature of attention, while $\phi^{(l)}$ needs to be $O(R)$-invariant to let $f^{(l)}$ invariant to orthogonal groups (Eq. 5). Likewise, we consider $g^{(l)}$ as a sequence-level propagation composed with an element-wise feedforward mapping: $g^{(l)} = g_{\mathsf{FFN}}^{(l)} \circ g_{\mathsf{Attn}}^{(l)}$, where $g_{\mathsf{Attn}}^{(l)}(\boldsymbol{X}, \boldsymbol{E}, \boldsymbol{M})_i = \sum_{j=1}^{N} B^{(l)}(\boldsymbol{X}, \boldsymbol{E}, \boldsymbol{M})_{ij} \boldsymbol{U}_V^{(l)} \boldsymbol{e}_j + \mathsf{Res}_{g_{\mathsf{Attn}}}^{(l)} \boldsymbol{e}_i$ performs pairwise aggregation where $\boldsymbol{U}_V^{(l)} \in \mathbb{R}^{ML \times ML}$, and the aggregation weights $B^{(l)}(\boldsymbol{X}, \boldsymbol{E}, \boldsymbol{M})_{ij} \in \mathbb{R}$ utilize both contextual and positional information, and $g_{\mathsf{FFN}}^{(l)}(\boldsymbol{X}, \boldsymbol{E})_i = \psi^{(l)}(\boldsymbol{x}_i) \boldsymbol{e}_i + \mathsf{Res}_{g_{\mathsf{FFN}}}^{(l)} \boldsymbol{e}_i$ modulates positional embeddings by token representations pointwisely with function $\psi^{(l)} : \mathbb{R}^C \to \mathbb{R}^{ML \times ML}$. To have $g$ satisfies Eqs. 4 and 5, $B^{(l)}$ needs to satisfies the following equation for every $\boldsymbol{P} \in \Pi(N)$ and $\boldsymbol{R} \in O(R)$: $B^{(l)}(\boldsymbol{P}\boldsymbol{X}, \boldsymbol{P}\boldsymbol{E}\boldsymbol{R}, \boldsymbol{P}\boldsymbol{M}\boldsymbol{P}^\top) = \boldsymbol{P}B^{(l)}(\boldsymbol{X}, \boldsymbol{E}, \boldsymbol{M}^\top)\boldsymbol{P}^\top$.

### B.1.2. $\Pi(5)$ GROUP WORD PROBLEM

The *group word problems* are a family of state tracking tasks, asking that, given a sequence of elements sampled from a generating set of a finite group, does it evaluate to the identity element after reduction? Formally, given a group $(G, \cdot)$ and its generating set $G^*$, we form a sequence $g_1, \cdots, g_N$ such that $g_i \in G^*$ for every $i \in [N]$. The decision problem is whether $g_1 \cdot ... \cdot g_N$ equals identity. The circuit complexity for this decision problem depends on the algebraic structure of the target group $G$. It is known that the the word problem with any non-solvable group is strictly NC[1]-complete (Immerman & Landau, 1995; Barrington, 1986). In particular, a typical example for non-solvable group is $\Pi(5)$, the permutation group over 5 entries (Merrill et al., 2024; Peng et al., 2025). We call this specific problem $\Pi(5)$ *group word problem*.

To this end, we focus on handling the sequences synthesized from $S(5)$ – a generating set of $\Pi(5)$. Note that $|S(5)| = 10$. Without loss of generality, we fix a canonical order in $S(5)$ and define the $n$-th element in $S(5)$ as $[\mathsf{src}(n) \leftrightarrow \mathsf{dst}(n)]$ (suppose $\mathsf{src}(n) < \mathsf{dst}(n)$) or $\boldsymbol{S}_n \in \mathbb{R}^{5 \times 5}$ in matrix form for $1 \leq n \leq 10$. Consider a vocabulary $V : [11] \to S(5) \cup \{[\mathsf{bos}]\}$, where $1, \cdots, 10$ are mapped to $S(5)$ and $11$ is mapped to the "beginning of sequence" token $[\mathsf{bos}]$. The input to the model

is a sequence $u = (u[1], \cdots, u[N]) \in [11]^N$, where we fix $u[1] \equiv 11$ (the [bos] token). The input sequence can be then illustrated as:

$$[\text{bos}], [\text{src}(u[2]) \leftrightarrow \text{dst}(u[2])], \cdots, [\text{src}(u[N]) \leftrightarrow \text{dst}(u[N])].$$

We denote the resultant permutation after reduction over the beginning $i$ transpositions as $\pi_i : [5] \rightarrow [5]$, which corresponds to the matrix form $\prod_{j=1}^{i} \boldsymbol{S}_{u[j]}$. The ground-truth labels are a boolean sequence $(y[1], \cdots, y[N]) \in \{0, 1\}^N$ such that $y[i] = 1$ if $\pi_i$ is identity, otherwise $y[i] = 0$. We call a model is capable of solving $\Pi(5)$ group word problem if for arbitrary input size $N$ and inputs $(u[1], \cdots, u[N])$, the model can precisely predict the ground-truth labels $(y[1], \cdots, y[N])$.

### B.1.3. MAIN RESULT AND PROOF

To show Theorem 4.1, it is equivalent to prove the following theorem, since $\Pi(5)$ group word problem is $\text{NC}^1$-complete, thus all the problems within $\text{NC}^1$ family can be reduced to under $\text{AC}^0$ reductions.

**Theorem B.1.** *There exists a constant-depth and constant-width transformer architecture from the model class defined in Sec. B.1.1, subject to permutation equivariance defined in Eq. 4 and orthogonal equivariance defined in Eq. 5, such that it is capable of solving $\Pi(5)$ group word problem, defined in Sec. B.1.2.*

*Proof.* Proof by the following construction.

**Embedding Layer.** We represent input sequence $u$ as a matrix of one-hot vectors: $\boldsymbol{X}^{(0)} \in \{0, 1\}^{N \times C}$ where $\boldsymbol{X}_i^{(0)} = \boldsymbol{\delta}_{u[i]} \in \mathbb{R}^{11}$ (i.e., $C = 11$). We initialize tensorial positional encoding $\boldsymbol{E}^{(0)} \in \mathbb{R}^{N \times C \times 6}$, i.e., $M = 1$ (thus omitted), $L = C, R = 6$, such that $\boldsymbol{E}_i^{(0)} = [\boldsymbol{\Xi}^\top, \boldsymbol{I}_{6 \times 6}]^\top$ for every $i \in [N]$, where $\boldsymbol{\Xi} \in \mathbb{R}^{C \times 6}$, $\boldsymbol{\Xi}_i = (\boldsymbol{\delta}_{\text{src}(i)} - \boldsymbol{\delta}_{\text{dst}(i)}) \in \mathbb{R}^6$ for $1 \le i \le 10$, and $\boldsymbol{\Xi}_{11} = \boldsymbol{0}_6$. In addition, we denote $\boldsymbol{M} \in \mathbb{R}^{N \times N}$ as the masking that describes the topology and dependencies among tokens. For word problems, $\boldsymbol{M}$ is a lower-triangular matrix, i.e., $\boldsymbol{M}_{ij} = 1$ when $j \le i$ while $\boldsymbol{M}_{ij} = 0$ when $j > i$.

**The First Layer.** We leverage the identity token mixer $f_{\text{Attn}}^{(1)} = f_{\text{Attn}_{\text{Id}}}$ and $f_{\text{FFN}}^{(1)} = f_{\text{FFN}_{\text{Id}}}$ as in Lemma B.4 such that $\boldsymbol{X}^{(1)} = \boldsymbol{X}^{(0)}$. For positional contextualization layers, we skip the sequence-level positional contextualizer, i.e., $g_{\text{Attn}}^{(1)} = \text{Id}$ by letting $B^{(1)} \equiv 0$ and $\text{Res}_{g_{\text{Attn}}}^{(1)} = 1$, while leveraging the point-wise positional contextualization with function

$$\psi^{(1)} : \boldsymbol{x} \mapsto \begin{bmatrix} \boldsymbol{x}^\top \\ \boldsymbol{0}_C^\top & \boldsymbol{0}_6^\top \\ & \boldsymbol{I}_{6 \times 6} \end{bmatrix} :$$

$$\boldsymbol{e}_i^{(1)} = g_{\text{FFN}}^{(1)}(\boldsymbol{X}^{(0)}, \boldsymbol{E}^{(0)})_i = \begin{bmatrix} \boldsymbol{x}_i^{(0)\top} \\ \boldsymbol{0}_C^\top & \boldsymbol{0}_6^\top \\ & \boldsymbol{I}_{6 \times 6} \end{bmatrix} \quad \boldsymbol{e}_i^{(0)} = \begin{bmatrix} \boldsymbol{x}_i^{(0)\top} \\ \boldsymbol{0}_C^\top & \boldsymbol{0}_6^\top \\ & \boldsymbol{I}_{6 \times 6} \end{bmatrix} \begin{bmatrix} \boldsymbol{\Xi} \\ \boldsymbol{I}_{6 \times 6} \end{bmatrix} = \begin{bmatrix} \boldsymbol{\Xi}_{u[i]}^\top \\ \boldsymbol{0}_6^\top \\ \boldsymbol{I}_{6 \times 6} \end{bmatrix} \in \mathbb{R}^{8 \times 6}.$$

Since the effective operations are only point-wise, permutation invariance in Eqs. 4 and 5 holds for this layer. Orthogonal group invariance can be also verified by seeing $\boldsymbol{e}_i^{(0)} \boldsymbol{R}$ gives $\boldsymbol{e}_i^{(1)} \boldsymbol{R}$ for every $\boldsymbol{R} \in O(6)$.

**The Second Layer.** We again leverage an identity attention block $f_{\text{Attn}}^{(2)} = f_{\text{Attn}_{\text{Id}}}$ (Lemma B.4) to process token features. Slightly different from identity feedforward layer $f_{\text{FFN}_{\text{Id}}}$, we intend to construct $f_{\text{FFN}}^{(2)}$ such that a new zero channel is concatenated to the inputs, i.e., $\boldsymbol{x}_i^{(2)} = [0 \quad \boldsymbol{x}_i^{(0)\top}]^\top \in \mathbb{R}^{C+1}$. We achieve this goal by setting the first linear layer as the identity matrix, activation function as ReLU, and the second linear layer as $[\boldsymbol{0}_C \quad \boldsymbol{I}_{C \times C}]^\top$. Note that ReLU does not change inputs here because all inputs are in the positive orthant.

Furthermore, we consider the following sequence-level contextualization $g_{\text{Attn}}^{(2)}$ with $\text{Res}_{g_{\text{Attn}}}^{(2)} = 1$ for positional embeddings. First, we compute pre-masking "attention" logits by "key" and "query" matrices $\boldsymbol{U}_K^{(2)} = \boldsymbol{U}_Q^{(2)} = \boldsymbol{\delta}_1^\top$ such that $\widetilde{B}^{(2)}(\boldsymbol{X}^{(1)}, \boldsymbol{E}^{(1)})_{ij} = (\boldsymbol{U}_Q^{(2)} \boldsymbol{e}_i^{(1)})(\boldsymbol{U}_K^{(2)} \boldsymbol{e}_j^{(1)})^\top = \boldsymbol{\Xi}_{u[i]}^\top \boldsymbol{\Xi}_{u[j]} \in \mathbb{R}$, and next we define $B^{(2)}(\boldsymbol{X}^{(1)}, \boldsymbol{E}^{(1)}, \boldsymbol{M})_{ij} = \left[\left(\boldsymbol{I}_{N \times N} + \boldsymbol{M} \odot \widetilde{B}^{(2)}(\boldsymbol{X}^{(1)}, \boldsymbol{E}^{(1)})\right)^{-1}\right]_{ij}$. We let $\boldsymbol{U}_V^{(2)} = [\boldsymbol{0}_6 \quad \boldsymbol{\delta}_1 \quad \boldsymbol{0}_{6 \times 6}]^\top$, and then the output of sequence-level

contextualization can be written as:

$$\widetilde{e}_i^{(2)} = \sum_{j=1}^N \left[ \left( I_{N \times N} + M \odot \widetilde{B}^{(2)}(X^{(1)}, E^{(2)}) \right)^{-1} \right]_{ij} \begin{bmatrix} \mathbf{0}_6^\top \\ \Xi_{u[j]}^\top \\ \mathbf{0}_{6 \times 6} \end{bmatrix} + \begin{bmatrix} \Xi_{u[i]}^\top \\ \mathbf{0}_6 \\ I_{6 \times 6} \end{bmatrix} = \begin{bmatrix} \Xi_{u[i]}^\top \\ \boldsymbol{\eta}_i^\top \\ I_{6 \times 6} \end{bmatrix} \in \mathbb{R}^{8 \times 6}.$$

where we define

$$\boldsymbol{\eta}_i = \sum_{j=1}^N \left[ \left( I_{N \times N} + M \odot \widetilde{B}^{(2)}(X^{(1)}, E^{(2)}) \right)^{-1} \right]_{ij} \Xi_{u[j]} \in \mathbb{R}^6.$$

We further adopt a feedforward contextualization $g_{\mathsf{FFN}}^{(2)}$ via $\psi^{(2)} : x \mapsto \left[ (\Xi^\top x)_1 \boldsymbol{\delta}_2 \quad \cdots \quad (\Xi^\top x)_6 \boldsymbol{\delta}_2 \quad \boldsymbol{\delta}_3 \quad \cdots \quad \boldsymbol{\delta}_8 \right]^\top$:

$$e_i^{(2)} = g_{\mathsf{FFN}}^{(2)}(X^{(1)}, \widetilde{E}^{(2)})_i = \begin{bmatrix} (\Xi^\top x)_1 \boldsymbol{\delta}_2^\top \\ \vdots \\ (\Xi^\top x)_6 \boldsymbol{\delta}_2^\top \\ \boldsymbol{\delta}_3^\top \\ \vdots \\ \boldsymbol{\delta}_8^\top \end{bmatrix} \begin{bmatrix} \Xi_{u[i]}^\top \\ \boldsymbol{\eta}_i^\top \\ I_{6 \times 6} \end{bmatrix} = \begin{bmatrix} \Xi^\top x_i^{(1)} \boldsymbol{\eta}_i^\top \\ I_{6 \times 6} \end{bmatrix} = \begin{bmatrix} \Xi_{u[i]} \boldsymbol{\eta}_i^\top \\ I_{6 \times 6} \end{bmatrix} \in \mathbb{R}^{12 \times 6}.$$

For the sake of coherence, we defer the justification of invariance for this layer to Lemma B.6.

**The Third Layer.** We leverage the weight configuration of Lemma B.5 to construct the attention layer: $f_{\mathsf{Attn}}^{(3)} = f_{\mathsf{Attn}_{\mathsf{Avg}}}$ and re-use the attention weights for cross-token positional contextualization (similar to Eq. 7): $B^{(3)}(X^{(2)}, E^{(2)}, M)_{ij} = A^{(3)}(X^{(2)}, E^{(2)}, M)_{ij} = \frac{1}{\|M_i\|_1}$. Further on, we let $U_V^{(3)} = I_{12 \times 12}$ and $W_V^{(3)} = \begin{bmatrix} 0 & \boldsymbol{\delta}_{11}^\top \\ \mathbf{0}_C & \mathbf{0}_{C \times C} \end{bmatrix}$. We enable residual connection for token-mixing attention $\mathsf{Res}_{f_{\mathsf{Attn}}}^{(3)} = 1$, while disabling it for cross-token contextualization $\mathsf{Res}_{g_{\mathsf{Attn}}}^{(3)} = 0$:

$$\widetilde{e}_i^{(3)} = \frac{1}{\|M_n\|_1} \sum_{j=1}^N M_{ij} U_V^{(3)} e_j^{(2)} = \frac{1}{\|M_i\|_1} \sum_{j=1}^N M_{ij} e_j^{(2)} = \frac{1}{\|M_i\|_1} \sum_{j=1}^N M_{ij} \begin{bmatrix} \Xi_{u[i]} \boldsymbol{\eta}_i^\top \\ I_{12 \times 6} \end{bmatrix} \in \mathbb{R}^{6 \times 6}, \tag{9}$$

$$\widetilde{x}_i^{(3)} = \frac{1}{\|M_i\|_1} \sum_{j=1}^N M_{ij} W_V^{(3)} x_j^{(2)} + x_i^{(2)} = \frac{1}{\|M_i\|_1} \sum_{j=1}^N M_{ij} \begin{bmatrix} \mathbb{1}\{x_j^{(0)} = \boldsymbol{\delta}_{11}\} \\ \mathbf{0}_C \end{bmatrix} + x_i^{(2)} \in \mathbb{R}^{C+1} \tag{10}$$

We note that for the word problem, $M$ is always a lower triangular matrix, and only the beginning token [bos] can equal 11, i.e., $u[1] = 11, u[i] \neq 11$ for all $i \geq 2$. In this scenario, we can express $\widetilde{E}^{(3)}$ and $\widetilde{X}^{(3)}$ as below:

$$\widetilde{e}_i^{(3)} = \begin{bmatrix} \frac{1}{i} \sum_{j=1}^i \Xi_{u[j]} \widetilde{e}_j^{(2)} \\ I_{6 \times 6} \end{bmatrix}, \quad \widetilde{x}_i^{(3)} = \begin{bmatrix} 1/i \\ x_i^{(0)} \end{bmatrix} \in \mathbb{R}^{C+1}. \tag{11}$$

Moving forward, we choose the identity feedforward mapping for element-wise token transformation: $f_{\mathsf{FFN}}^{(3)} = f_{\mathsf{FFN}_{\mathsf{Id}}}$, i.e., $x_i^{(3)} = [1/i \quad x_i^{(0)\top}]^\top$. The element-wise positional contextualization $g_{\mathsf{FFN}}^{(3)}$ is given by a nonlinear transformation $\psi^{(3)} : x \mapsto [-(\boldsymbol{\delta}_1^\top x)^{-1} I_{6 \times 6} \quad I_{6 \times 6}]$:

$$e_i^{(3)} = g_{\mathsf{FFN}}^{(3)}(\widetilde{X}^{(2)}, \widetilde{E}^{(2)})_i = \begin{bmatrix} -(\boldsymbol{\delta}_1^\top x)^{-1} I_{6 \times 6} & I_{6 \times 6} \end{bmatrix} \begin{bmatrix} \frac{1}{i} \sum_{j=1}^i \Xi_{u[j]} \boldsymbol{\eta}_j^\top \\ I_{6 \times 6} \end{bmatrix} = I_{6 \times 6} - \sum_{j=1}^i \Xi_{u[j]} \boldsymbol{\eta}_j^\top, \tag{12}$$

where the second equality holds for the word problem case. By Lemma B.5, we show that Eqs. 9 and 10 satisfy permutation equivariance (Eq. 4). Cross-token contextualization (Eq. 9) is orthogonal equivariant because $B^{(3)}$ is invariant to (or even independent of) $E^{(2)}$, and Eq. 9 is a linear combination of $e_j^{(2)}$'s. The point-wise contextualization is a left matrix multiplication on $\widetilde{E}^{(2)}$ (Eq. 12), preserving orthogonal equivariance as well.

Significantly, by Lemma B.7, we show that the transpose of Eq. 12 is a cumulative Householder product (Grazzi et al., 2024; Yang et al., 2024; 2025):

$$e_i^{(3)\top} = \prod_{j=1}^{i} \left( I_{6\times 6} - \Xi_{u[j]} \Xi_{u[j]}^{\top} \right) = \prod_{j=1}^{i} \begin{bmatrix} S_{u[j]} & 0 \\ 0 & 1 \end{bmatrix} = \begin{bmatrix} \prod_{j=1}^{i} S_{u[j]} & 0 \\ 0 & 1 \end{bmatrix},$$

where the second equality is by Lemma B.8 (Peng et al., 2023a).

**The Fourth Layer.** We follow the proof of Yang et al. (2025), Theorem 2.1 to construct the output layer. For the attention, we define weight matrices for $f_{\text{Attn}}^{(4)}$:

$$W_K^{(4)} = \begin{bmatrix} 0_{6\times 10} & \delta_1 + 2\delta_2 + 3\delta_3 + 4\delta_4 + 5\delta_5 - \delta_6 \end{bmatrix} \begin{bmatrix} 0_C & I_{C\times C} \end{bmatrix} \in \mathbb{R}^{6\times(C+1)},$$

$$W_Q^{(4)} = (\delta_1 + 2\delta_2 + 3\delta_3 + 4\delta_4 + 5\delta_5 + 54.5\delta_6)1_{11}^{\top} \begin{bmatrix} 0_C & I_{C\times C} \end{bmatrix} \in \mathbb{R}^{6\times(C+1)},$$

$$W_V^{(4)} = \begin{bmatrix} 0_{(C+1)\times 10} & -\delta_1 \end{bmatrix} \begin{bmatrix} 0_C & I_{C\times C} \end{bmatrix} \in \mathbb{R}^{(C+1)\times(C+1)},$$

where $\begin{bmatrix} 0_C & I_{C\times C} \end{bmatrix}$ is applied to extract the lower block of $x_i^{(3)}$: $\begin{bmatrix} 0_C & I_{C\times C} \end{bmatrix} [1/i \quad x_i^{(0)\top}]^{\top} = x_i^{(0)}$. We also utilize the positional encoding via $\phi^{(4)}$ defined as below:

$$\phi^{(4)}(e_i^{(3)}, e_j^{(3)}) = e_j^{(3)} e_i^{(3)\top} \in \mathbb{R}^{6\times 6}.$$

First of all, $\phi^{(4)}$ is invariant to orthogonal transformations on $\{e_i^{(3)}\}_{i\in[N]}$ due to its inner-product form, so is $f_{\text{Attn}}^{(4)}$. Next, we analyze the pre-softmax attention logits $\widetilde{A}^{(4)}(X^{(3)}, E^{(3)}, M)_{ij} = (W_Q x_i^{(3)})^{\top} \phi^{(4)}(e_i^{(3)}, e_j^{(3)})(W_K x_j^{(3)})$. For every $i \in [N]$ and $j \neq 1$, $\widetilde{A}^{(4)}(X^{(3)}, E^{(3)}, M)_{ij} = 0$ because $W_K x_j^{(3)} = 0_6$. For $j = 1$:

$$\widetilde{A}^{(4)}(X^{(3)}, E^{(3)}, M)_{N,1} = \begin{bmatrix} 1 \\ 2 \\ 3 \\ 4 \\ 5 \\ 54.5 \end{bmatrix}^{\top} \begin{bmatrix} \prod_{j=1}^{i} S_{u[j]} & 0 \\ 0 & 1 \end{bmatrix} \begin{bmatrix} 1 \\ 2 \\ 3 \\ 4 \\ 5 \\ -1 \end{bmatrix} = \sum_{v=1}^{5} v\pi_i(v) - 54.5,$$

where we use the fact that $e_1^{(3)} = I_{6\times 6}$ and $x_1^{(3)} = [1 \quad \delta_{11}^{\top}]^{\top}$. Observe that $\sum_{v=1}^{5} v\pi_i(v) \leq 55$ with equality hold if and only if $\pi_i(v) = v$ for every $v \in [5]$, i.e., $\pi_i$ is the identity permutation. This is, $\widetilde{A}^{(4)}(X^{(3)}, E^{(3)}, M)_{i,1} \geq 0.5$ if $\pi_i$ is the identity, and $\widetilde{A}^{(4)}(X^{(3)}, E^{(3)}, M)_{i,1} \leq -0.5$ otherwise. After softmax normalization, followed by applying the attention, value matrix, and skip connection (i.e., $\text{Res}_{f_{\text{Attn}}}^{(4)} = 1$), the output of this attention layer becomes:

$$\widetilde{x}_i^{(4)} = \sum_{j=1}^{i} A^{(4)}(X^{(3)}, E^{(3)}, M)_{i,j} W_V^{(4)} x_j^{(3)} + x_i^{(3)} = -A^{(4)}(X^{(3)}, E^{(3)}, M)_{i,1}\delta_1 + x_i^{(3)}$$

$$= \begin{bmatrix} -\frac{\exp(\widetilde{A}^{(4)}(X^{(3)}, E^{(3)}, M)_{i,1})}{\exp(\widetilde{A}^{(4)}(X^{(3)}, E^{(3)}, M)_{i,1})+i-1} + \frac{1}{i} \\ x_i^{(0)} \end{bmatrix} \in \mathbb{R}^{C+1}.$$

One can see that if $\pi_i$ is identity, then $\widetilde{A}^{(4)}(X^{(3)}, E^{(3)}, M)_{i,1}) \geq 0$, and $\widetilde{x}_{i,1}^{(4)} \leq 0$. Otherwise, $\widetilde{A}^{(4)}(X^{(3)}, E^{(3)}, M)_{i,1}) < 0$, and $\widetilde{x}_{i,1}^{(4)} > 0$. By this observation, we construct $f_{\text{FFN}}^{(4)}$ to read the first element from $\widetilde{x}_i^{(4)}$ and map it to the corresponding label via a two-layer MLP: we adopt $-\delta_1$ as the first linear layer, hard sigmoid $\min(1, \max(0, \cdot))$ as the non-linear activation, and identity as the second layer. The final output can be analytically written as:

$$x_i^{(4)} = \text{HardSigmoid}(-\delta_1^{\top})\widetilde{x}_i^{(4)} = \begin{cases} 1 & \text{if } \pi_i \text{ is identity} \\ 0 & \text{otherwise} \end{cases},$$

which concludes the proof. □

*Remark* B.2. Different from the construction of Yang et al. (2025), Theorem 2.1, we do not encode the sequence length in the query matrix. In fact, if the construction is length dependent, the architecture could potentially lose uniformity of the target circuit family when input size is unbounded. We note that the length dependency in the construction of Yang et al. (2025), Theorem 2.1 is likely due to the consideration of the computational precision of attention. Under the similar setting, it is necessary to modify $\psi^{(3)}$ as the function: $\boldsymbol{x} \mapsto [-(\boldsymbol{\delta}_1^\top \boldsymbol{x})^{-2} \boldsymbol{I}_{6\times 6} \quad \boldsymbol{I}_{6\times 6}]$ to scale the pre-softmax attention logit by $n$.

*Remark* B.3. We emphasize that our proof is constructive and does not rely on the universal approximation property of Attention (Yun et al., 2019) or MLPs (Yarotsky, 2022). As a result, the network has constant width and depth and represents a precise classifier solving the $\Pi(5)$ word problem.

### B.1.4. AUXILIARY RESULTS

**Lemma B.4.** *(Identity mappings.) There exists an attention layer $f_{\mathsf{Attn_{Id}}}$ and feedforward layer $f_{\mathsf{FFN_{Id}}}$ that meets Eq. 4 and Eq. 5 such that $f_{\mathsf{Attn_{Id}}}(\boldsymbol{X}, \boldsymbol{E}, \boldsymbol{M}) = \boldsymbol{X}$ and $f_{\mathsf{FFN_{Id}}}(\boldsymbol{X}) = \boldsymbol{X}$ for every inputs $\boldsymbol{X}, \boldsymbol{E}, \boldsymbol{M}$.*

*Proof.* Proof by construction. For the attention, we let $\boldsymbol{W}_V = \boldsymbol{0}_{C\times C}$ in attention and $\mathsf{Res}_{f_{\mathsf{Attn_{Id}}}} = 1$. For the feedforward layer, we choose the associated $\mathsf{FFN_{Id}}(\cdot)$ as a two-layer MLP with the last weight matrix all zeros, and let $\mathsf{Res}_{f_{\mathsf{FFN_{Id}}}} = 1$. Eq. 4 and Eq. 5 hold automatically. $\qquad\square$

**Lemma B.5.** *(Neighborhood mean.) There exists an attention layer $f_{\mathsf{Attn_{Avg}}}$ with free value matrix $\boldsymbol{W}_V$ that meets Eq. 4 and Eq. 5 such that $f_{\mathsf{Attn_{Avg}}}(\boldsymbol{X}, \boldsymbol{E}, \boldsymbol{M})_i = (\sum_{j=1}^N \boldsymbol{M}_{i,j}\boldsymbol{W}_V \boldsymbol{x}_j)/(\sum_{j'=1}^N \boldsymbol{M}_{i,j'})$.*

*Proof.* Proved by letting $\boldsymbol{W}_Q$ and $\boldsymbol{W}_K$ all zero and disable the residual connection $\mathsf{Res}_{f_{\mathsf{Attn_{Avg}}}} = 0$. Permutation equivariance are satisfied automatically due to the nature of masked attention. To be more specific, for every permutation matrix $\boldsymbol{P} \in \Pi(N)$ and its corresponding permutation map: $\pi : [N] \to [N]$:

$$f_{\mathsf{Attn_{Avg}}}(\boldsymbol{PX}, \boldsymbol{PE}, \boldsymbol{PMP}^\top)_i = \frac{\sum_{j=1}^N \boldsymbol{M}_{\pi(i),\pi(j)}\boldsymbol{W}_V \boldsymbol{x}_{\pi(j)}}{\sum_{j'=1}^N \boldsymbol{M}_{\pi(i),\pi(j')}} = \frac{\sum_{j=1}^N \boldsymbol{M}_{\pi(i),j}\boldsymbol{W}_V \boldsymbol{x}_j}{\sum_{j'=1}^N \boldsymbol{M}_{\pi(i),j'}} = f_{\mathsf{Attn_{Avg}}}(\boldsymbol{X}, \boldsymbol{E}, \boldsymbol{M})_{\pi(i)},$$

where the second equality is because exchanging terms in summation does not change the outcome. Hereby, we have shown $f_{\mathsf{Attn_{Avg}}}(\boldsymbol{PX}, \boldsymbol{PE}, \boldsymbol{PMP}^\top) = \boldsymbol{P} f_{\mathsf{Attn_{Avg}}}(\boldsymbol{X}, \boldsymbol{E}, \boldsymbol{M})$. Orthogonal invariance is given by the independence of $\boldsymbol{E}$. $\qquad\square$

**Lemma B.6.** *The second positional contextualization layers $g_{\mathsf{Attn}}^{(2)}$ and $g_{\mathsf{FFN}}^{(2)}$ in Theorem B.1 satisfy permutation equivariance (Eq. 4) and orthogonal equivariance (Eq. 5).*

*Proof.* First of all, $g_{\mathsf{FFN}}^{(2)}(\boldsymbol{X}, \boldsymbol{E}, \boldsymbol{M})$ is element-wise, thus satisfies permutation equivariance (Eq. 4). Also, $g_{\mathsf{FFN}}^{(2)}(\boldsymbol{X}, \boldsymbol{E}, \boldsymbol{M})$ is a left matrix multiplication on every positional encoding $\boldsymbol{e}_i$, hereby, orthogonal equivariant (Eq. 5). It remains to demonstrate the two invariances for $g_{\mathsf{Attn}}^{(2)}$. For every $\boldsymbol{P} \in \Pi(N)$ and its corresponding permutation map $\pi : [N] \to [N]$, we have that:

$$\widetilde{B}^{(2)}(\boldsymbol{PX}, \boldsymbol{PE})_{ij} = \boldsymbol{e}_{\pi(i)}\boldsymbol{e}_{\pi(j)}^\top = \widetilde{B}^{(2)}(\boldsymbol{X}, \boldsymbol{E})_{\pi(i),\pi(j)},$$

which indicates $\widetilde{B}^{(2)}(\boldsymbol{PX}, \boldsymbol{PE}) = \boldsymbol{P}\widetilde{B}^{(2)}(\boldsymbol{X}, \boldsymbol{E})\boldsymbol{P}^\top$. Next we analyze $B^{(2)}(\boldsymbol{X}, \boldsymbol{E}, \boldsymbol{M})$:

$$\begin{aligned}
B^{(2)}(\boldsymbol{PX}, \boldsymbol{PE}, \boldsymbol{PMP}^\top) &= \left(\boldsymbol{I} + \boldsymbol{PMP}^\top \odot \widetilde{B}^{(2)}(\boldsymbol{PX}, \boldsymbol{PE})\right)^{-1} = \left(\boldsymbol{I} + \boldsymbol{PMP}^\top \odot \boldsymbol{P}\widetilde{B}^{(2)}(\boldsymbol{X}, \boldsymbol{E})\boldsymbol{P}^\top\right)^{-1} \\
&= \left(\boldsymbol{I} + \boldsymbol{P}\left(\boldsymbol{M} \odot \widetilde{B}^{(2)}(\boldsymbol{X}, \boldsymbol{E})\right)\boldsymbol{P}^\top\right)^{-1} = \left(\boldsymbol{P}\left(\boldsymbol{I} + \boldsymbol{M} \odot \widetilde{B}^{(2)}(\boldsymbol{X}, \boldsymbol{E})\right)\boldsymbol{P}^\top\right)^{-1} \\
&= \boldsymbol{P}\left(\boldsymbol{I} + \boldsymbol{M} \odot \widetilde{B}^{(2)}(\boldsymbol{X}, \boldsymbol{E})\right)^{-1}\boldsymbol{P}^\top.
\end{aligned}$$

By above derivation, we have:

$$
\begin{aligned}
g_{\text{Attn}}^{(2)}(\boldsymbol{PX}, \boldsymbol{PE}, \boldsymbol{PMP}^\top)_i &= \sum_{j=1}^{N} B^{(2)}(\boldsymbol{PX}, \boldsymbol{PE}, \boldsymbol{PMP}^\top)_{ij} \boldsymbol{e}_{\pi(j)} = \sum_{j=1}^{N} \left[ \boldsymbol{P} B^{(2)}(\boldsymbol{X}, \boldsymbol{E}, \boldsymbol{M}) \boldsymbol{P}^\top \right]_{ij} \boldsymbol{e}_{\pi(j)} \\
&= \sum_{j=1}^{N} B^{(2)}(\boldsymbol{X}, \boldsymbol{E}, \boldsymbol{M})_{\pi(i), \pi(j)} \boldsymbol{e}_{\pi(j)} = \sum_{j=1}^{N} B^{(2)}(\boldsymbol{X}, \boldsymbol{E}, \boldsymbol{M})_{\pi(i), j} \boldsymbol{e}_j \\
&= g_{\text{Attn}}^{(2)}(\boldsymbol{X}, \boldsymbol{E}, \boldsymbol{M})_{\pi(i)},
\end{aligned}
$$

which proves the permutation equivariance: $g_{\text{Attn}}^{(2)}(\boldsymbol{PX}, \boldsymbol{PE}, \boldsymbol{PMP}^\top) = \boldsymbol{P} g_{\text{Attn}}^{(2)}(\boldsymbol{X}, \boldsymbol{E}, \boldsymbol{M})$. To show orthogonal equivariance, we note that, for every $\boldsymbol{R} \in O(R)$:

$$
\widetilde{B}^{(2)}(\boldsymbol{X}, \boldsymbol{ER})_{ij} = \boldsymbol{e}_i^{(1)} \boldsymbol{R} \boldsymbol{R}^\top \boldsymbol{e}_j^\top = \boldsymbol{e}_i \boldsymbol{e}_j^\top = \widetilde{B}^{(2)}(\boldsymbol{X}, \boldsymbol{E})_{ij}.
$$

This implies $B^{(2)}(\boldsymbol{X}, \boldsymbol{ER}) = B^{(2)}(\boldsymbol{X}, \boldsymbol{E})$ and henceforth,

$$
g_{\text{Attn}}^{(2)}(\boldsymbol{X}, \boldsymbol{ER}, \boldsymbol{M})_i = \sum_{j=1}^{N} B^{(2)}(\boldsymbol{X}, \boldsymbol{ER}, \boldsymbol{M})_{ij} \boldsymbol{e}_j \boldsymbol{R} = \sum_{j=1}^{N} B^{(2)}(\boldsymbol{X}, \boldsymbol{E}, \boldsymbol{M})_{ij} \boldsymbol{e}_j \boldsymbol{R} = g_{\text{Attn}}^{(2)}(\boldsymbol{X}, \boldsymbol{E}, \boldsymbol{M})_i \boldsymbol{R},
$$

which shows the orthogonal equivariance. $\qquad\square$

**Lemma B.7** (Bischof & Van Loan (1987); Yang et al. (2024)). *Consider a sequence of vectors $\boldsymbol{\Xi} = [\boldsymbol{\xi}_1, \cdots, \boldsymbol{\xi}_N]^\top \in \mathbb{R}^{N \times R}$, the following equation holds for every $i \in [N]$:*

$$
\prod_{j=1}^{i} \left( \boldsymbol{I}_{R \times R} - \boldsymbol{\xi}_j \boldsymbol{\xi}_j^\top \right) = \boldsymbol{I}_{R \times R} - \sum_{j=1}^{i} \left[ \left( \boldsymbol{I}_{N \times N} + \boldsymbol{M} \odot \boldsymbol{\Xi} \boldsymbol{\Xi}^\top \right)^{-1} \boldsymbol{\Xi} \right]_j \boldsymbol{\xi}_j^\top,
$$

*where $\boldsymbol{M}$ is a lower-triangular mask matrix, i.e., $\boldsymbol{M}_{ij} = 1$ if $j \leq i$, otherwise $\boldsymbol{M}_{ij} = 0$.*

*Proof.* The proof follows from the standard derivation of WY representation for accumulating Householder matrices (Bischof & Van Loan, 1987). A clear derivation can be seen in Yang et al. (2024). First of all, let $\boldsymbol{w}_i = \boldsymbol{\xi}_i - \sum_{j=1}^{i-1} (\boldsymbol{\xi}_j^\top \boldsymbol{\xi}_i) \boldsymbol{w}_j \in \mathbb{R}^R$ for $i \geq 2$ and $\boldsymbol{w}_1 = \boldsymbol{\xi}_1$. We let $\boldsymbol{W} = [\boldsymbol{w}_1, \cdots, \boldsymbol{w}_N]^\top \in \mathbb{R}^{N \times R}$. Then expressing $\boldsymbol{W}$ in matrix form:

$$
\boldsymbol{W} = \boldsymbol{\Xi} - \left( \boldsymbol{M} \odot \boldsymbol{\Xi} \boldsymbol{\Xi}^\top \right) \boldsymbol{W},
$$

which gives $\boldsymbol{W} = (\boldsymbol{I}_{N \times N} + \boldsymbol{M} \odot \boldsymbol{\Xi} \boldsymbol{\Xi}^\top)^{-1} \boldsymbol{\Xi}$. We denote $\boldsymbol{P}_i = \prod_{j=1}^{i} (\boldsymbol{I}_{R \times R} - \boldsymbol{\xi}_j \boldsymbol{\xi}_j^\top)$ and perform the following induction. The induction hypothesis is $\boldsymbol{P}_i = \boldsymbol{I}_{R \times R} - \sum_{j=1}^{i} \boldsymbol{w}_j \boldsymbol{\xi}_j^\top$. Obviously, $\boldsymbol{P}_1 = \boldsymbol{I}_{R \times R} - \boldsymbol{\xi}_1 \boldsymbol{\xi}_1^\top = \boldsymbol{I}_{R \times R} - \boldsymbol{w}_1 \boldsymbol{\xi}_1^\top$. Then for $i \geq 2$, we can derive that:

$$
\begin{aligned}
\boldsymbol{P}_i &= \prod_{j=1}^{i} \left( \boldsymbol{I}_{R \times R} - \boldsymbol{\xi}_j \boldsymbol{\xi}_j^\top \right) = \boldsymbol{P}_{i-1} \left( \boldsymbol{I}_{R \times R} - \boldsymbol{\xi}_i \boldsymbol{\xi}_i^\top \right) = \left( \boldsymbol{I}_{R \times R} - \sum_{j=1}^{i-1} \boldsymbol{w}_j \boldsymbol{\xi}_j^\top \right) \left( \boldsymbol{I}_{R \times R} - \boldsymbol{\xi}_i \boldsymbol{\xi}_i^\top \right) \\
&= \boldsymbol{I}_{R \times R} - \sum_{j=1}^{i-1} \boldsymbol{w}_j \boldsymbol{\xi}_j^\top - \boldsymbol{\xi}_i \boldsymbol{\xi}_i^\top + \sum_{j=1}^{i-1} \boldsymbol{w}_j \boldsymbol{\xi}_j^\top \boldsymbol{\xi}_i \boldsymbol{\xi}_i^\top = \boldsymbol{I}_{R \times R} - \sum_{j=1}^{i-1} \boldsymbol{w}_j \boldsymbol{\xi}_j^\top - \left( \boldsymbol{\xi}_i - \sum_{j=1}^{i-1} (\boldsymbol{\xi}_j^\top \boldsymbol{\xi}_i) \boldsymbol{w}_j \right) \boldsymbol{\xi}_i^\top \\
&= \boldsymbol{I}_{R \times R} - \sum_{j=1}^{i-1} \boldsymbol{w}_j \boldsymbol{\xi}_j^\top - \boldsymbol{w}_i \boldsymbol{\xi}_i^\top = \boldsymbol{I}_{R \times R} - \sum_{j=1}^{i} \boldsymbol{w}_j \boldsymbol{\xi}_j^\top.
\end{aligned}
$$

which justifies the hypothesis. We completes the proof by writing $\boldsymbol{w}_i$ in the matrix form:

$$
\boldsymbol{P}_i = \boldsymbol{I}_{R \times R} - \sum_{j=1}^{i} \left[ \left( \boldsymbol{I}_{N \times N} + \boldsymbol{M} \odot \boldsymbol{\Xi} \boldsymbol{\Xi}^\top \right)^{-1} \boldsymbol{\Xi} \right]_j \boldsymbol{\xi}_j^\top, \tag{13}
$$

as desired. $\qquad\square$

**Lemma B.8** ([Peng et al. (2025)](#)). *Given a swap group of $n$ elements $S(n)$, for every swap $[i \leftrightarrow j]$, its matrix representation $\boldsymbol{S} \in S(n)$ can be written as $\boldsymbol{S} = \boldsymbol{I}_n - (\boldsymbol{\delta}_i - \boldsymbol{\delta}_j)(\boldsymbol{\delta}_i - \boldsymbol{\delta}_j)^\top$.*

*Proof.* Proof by noticing that $\boldsymbol{S}_i = \boldsymbol{\delta}_j$, $\boldsymbol{S}_j = \boldsymbol{\delta}_i$, and $\boldsymbol{S}_k = \boldsymbol{\delta}_k$ for every $k \neq i$ and $k \neq j$. □

### B.2. Formal Statement and Proof for Proposition 4.2

We first formulate a general computational paradigm of transformers with TAPE.

**Inputs.** Let $\boldsymbol{X}^{(0)} \in \mathbb{R}^{N \times C}$ be the input sequence of token embeddings. We consider two types of initialization of positional encoding:

- *RoPE.* Construct positional embeddings as a tensor: $\boldsymbol{E}^{(0)} \in \mathbb{R}^{N \times C/2 \times 2 \times 2}$. For every $i \in [N], m \in [C/2]$, we let
$$\boldsymbol{e}_{i,m}^{(0)} = \begin{bmatrix} \cos(\theta_m i) & -\sin(\theta_m i) \\ \sin(\theta_m i) & \cos(\theta_m i) \end{bmatrix}, \text{ where } \theta_m = -10000^{2m/C}.$$

- *Reweighted Random Fourier Features.* Construct positional embeddings as a tensor: $\boldsymbol{E}^{(0)} \in \mathbb{R}^{N \times M \times L \times R}$. For every $i \in [N], m \in [M], l \in [L]$

$$\boldsymbol{e}_{i,m,l}^{(0)} = \sqrt{\frac{2}{R}} \begin{bmatrix} \cdots & w_{m,l,r}\cos(\theta_{m,r}i) & w_{m,l,r}\sin(\theta_{m,r}i) & \cdots \end{bmatrix}^\top, r \in [R/2]$$

where $\{\theta_{m,r}\}_{m\in[M],r\in[R/2]}$ are often chosen as (quasi-)Monte-Carlo samples from a distribution, and $\{w_{m,l,r}\}_{m\in[M],l\in[L],r\in[R/2]}$ are a collection of coefficients.

**Model.** We consider a transformer $F : \mathbb{R}^{N \times C} \times \mathbb{R}^{N \times M \times L \times R} \to \mathbb{R}^{N \times C}$ consisting of $T$ transformer blocks. For the $t$-th block, we consider it employs function $f^{(t)} : \mathbb{R}^{N \times C} \times \mathbb{R}^{N \times M \times L \times R} \to \mathbb{R}^{N \times C}$ to update features, and function $g^{(t)} : \mathbb{R}^{N \times C} \times \mathbb{R}^{N \times M \times L \times R} \to \mathbb{R}^{N \times M \times L \times R}$ to update positional embeddings:

$$\boldsymbol{X}^{(t)} = f^{(t)}(\boldsymbol{X}^{(t-1)}, \boldsymbol{E}^{(t-1)}), \quad \boldsymbol{E}^{(t)} = g^{(t)}(\boldsymbol{X}^{(t-1)}, \boldsymbol{E}^{(t-1)}), \quad t \in [T].$$

We denote $\boldsymbol{X}^{(T)}$ as the final output of $F(\boldsymbol{X}^{(0)}, \boldsymbol{E}^{(0)})$. We assume $f^{(t)}$ and $g^{(t)}$ jointly satisfies our invariance properties Eqs. 4 and 5.

**Phase Shift.** We say a transformer is invariant to translation if the final outputs $\boldsymbol{X}^{(T)}$ remains unchanged when the input token indices are shifted by an offset. This implies the attention mechanism inside depends on positional information based on the relative distances instead of absolute indices. Formally, when an offset $\Delta \in \mathbb{R}$ is applied to all positions, the initial positional embeddings undergo a phase shift. We denote the positional embeddings with translation $\Delta$ as $\widetilde{\boldsymbol{E}}^{(0)}$, whose per-token representations $\{\widetilde{\boldsymbol{e}}_i^{(0)}\}_{i\in[N]}$ can be written as:

- *RoPE.* $\widetilde{\boldsymbol{e}}_{i,m}^{(0)} = \begin{bmatrix} \cos(\theta_m(i+\Delta)) & -\sin(\theta_m(i+\Delta)) \\ \sin(\theta_m(i+\Delta)) & \cos(\theta_m(i+\Delta)) \end{bmatrix}$ for every $i \in [N], m \in [C/2]$.

- *Random Fourier Features.* For every $i \in [N], m \in [M], l \in [L]$

$$\widetilde{\boldsymbol{e}}_{i,m,l}^{(0)} = \sqrt{\frac{2}{R}} \begin{bmatrix} \cdots & w_{m,l,r}\cos(\theta_{m,r}(i+\Delta)) & w_{m,l,r}\sin(\theta_{m,r}(i+\Delta)) & \cdots \end{bmatrix}^\top.$$

We denote the intermediate outputs yielded by shifted positional embeddings as $(\widetilde{\boldsymbol{X}}^{(t)}, \widetilde{\boldsymbol{E}}^{(t)})$, for every $t \in [T]$.

**Main Result.** We provide a formal version and proof of Proposition 4.2 as below:

**Proposition B.9** (Formal version of Proposition 4.2)**.** *Assume $f^{(t)}$ and $g^{(t)}$ satisfies Eqs. 4 and 5 for every $t \in [T]$. Then for any shift $\Delta \in \mathbb{R}$, we have $F(\boldsymbol{X}^{(0)}, \boldsymbol{E}^{(0)}) = F(\boldsymbol{X}^{(0)}, \widetilde{\boldsymbol{E}}^{(0)})$.*

*Proof.* First, we observe that a shift on the token indices translates to an orthogonal transformation on the embedding space for both RoPE and random Fourier features. For RoPE, this can be seen by:

$$\widetilde{\boldsymbol{e}}_{i,m}^{(0)} = \begin{bmatrix} \cos(\theta_m i) & -\sin(\theta_m i) \\ \sin(\theta_m i) & \cos(\theta_m i) \end{bmatrix} \underbrace{\begin{bmatrix} \cos(\theta_m \Delta) & -\sin(\theta_m \Delta) \\ \sin(\theta_m \Delta) & \cos(\theta_m \Delta) \end{bmatrix}}_{\boldsymbol{O}_{RoPE,\Delta,m}},$$

where the extracted matrix $\boldsymbol{O}_{RoPE,\Delta,m}$ is an orthogonal matrix. For random Fourier features, we observe that $\widetilde{\boldsymbol{e}}_{i,m} = \boldsymbol{e}_{i,m}^{(0)} \boldsymbol{O}_{RFF,\Delta,m}$, where $\boldsymbol{O}_{RFF,\Delta,m} = \text{diag}(\boldsymbol{O}_{RFF,\Delta,m,1}, \cdots \boldsymbol{O}_{RFF,\Delta,m,R/2})$, and each $\boldsymbol{O}_{RFF,\Delta,r} \in \mathbb{R}^{2 \times 2}$ is defined as:

$$\boldsymbol{O}_{RFF,\Delta,m,r} = \begin{bmatrix} \cos(\theta_{m,r}\Delta) & -\sin(\theta_{m,r}\Delta) \\ \sin(\theta_{m,r}\Delta) & \cos(\theta_{m,r}\Delta) \end{bmatrix},$$

which shows the orthogonality of $\boldsymbol{O}_{RFF,\Delta,m}$.

Now we apply the orthogonal invariance in an inductive argument. We make hypothesis that $\boldsymbol{X}^{(t)} = \widetilde{\boldsymbol{X}}^{(t)}$ and $\boldsymbol{E}^{(t)} = \widetilde{\boldsymbol{E}}^{(t)} \boldsymbol{O}$ for some $t \in [T] \cup \{0\}$ with $\boldsymbol{O}$ corresponding to the initialization specific orthogonal transformation. It is obvious that this holds for $t = 0$. Then by the orthogonal invariance and equivariant of $f^{(t+1)}$ and $g^{(t+1)}$, we have that for every $t \in [T-1] \cup \{0\}$:

$$\widetilde{\boldsymbol{X}}^{(t+1)} = f^{(t+1)}(\widetilde{\boldsymbol{X}}^{(t)}, \widetilde{\boldsymbol{E}}^{(t)}) = f^{(t+1)}(\boldsymbol{X}^{(t)}, \boldsymbol{E}^{(t)}\boldsymbol{O}) = f^{(t+1)}(\boldsymbol{X}^{(t)}, \boldsymbol{E}^{(t)}) = \boldsymbol{X}^{(t+1)}$$

$$\widetilde{\boldsymbol{E}}^{(t+1)} = g^{(t+1)}(\widetilde{\boldsymbol{X}}^{(t)}, \widetilde{\boldsymbol{E}}^{(t)}) = g^{(t+1)}(\boldsymbol{X}^{(t)}, \boldsymbol{E}^{(t)}\boldsymbol{O}) = g^{(t+1)}(\boldsymbol{X}^{(t)}, \boldsymbol{E}^{(t)})\boldsymbol{O} = \boldsymbol{E}^{(t+1)}\boldsymbol{O},$$

which concludes the proof by induction. $\square$

## C. Implementation Details

**Experiment Settings.** In Sec.5.1, the model architecture includes 16 layers, a hidden dimension of 1024, an intermediate dimension of 2048, and 16 attention heads, resulting in approximately 120M parameters. In Sec.5.2, the architecture features 12 layers, a hidden dimension of 768, an intermediate dimension of 3072, and 12 attention heads, totaling approximately 155M parameters. The training recipe in three experiments are presented in Tab. 5.

*Table 5.* Training recipe for language model pre-training and fine-tuning in experiments.

| | Arithmetic (§5.1) | C4 Pre-training (§5.2) | SCROLLS (§5.2) | Context Extension (§5.3) |
|---|---|---|---|---|
| Sequence length | 40 + 40 | 1024 | 1024 | 8096 |
| Batch size | 512 | 512 | 64 | 64 |
| Number of iterations | 20k | 10k | 1k | 1k |
| Attention dropout prob. | 0.0 | 0.0 | 0.0 | 0.0 |
| Optimizer | AdamW | AdamW | AdamW | AdamW |
| Learning rate | $1 \times 10^{-4}$ | $1 \times 10^{-4}$ | $1 \times 10^{-5}$ | $2 \times 10^{-5}$ |

**Masked Multi-Head Mechanism.** The masked multi-head attention is a key design in the original Transformer and is well compatible with our method. To enforce causality in language generation, the Transformer masks out (sets to $-\infty$) all values in the input to the softmax that correspond to illegal connections from future tokens to current tokens. This is similarly implemented in our enhanced Transformer for language modeling. To allow the model to jointly attend to information from different representation subspaces at different positions, multiple attention outputs are computed in parallel with multiple attention heads, and then mixed through concatenation and a linear transformation. In our enhanced Transformer, the head dimension is added to both token embeddings and positional embeddings, resulting in $\boldsymbol{X} \in \mathbb{R}^{N \times H \times M \times B}$ and $\boldsymbol{E} \in \mathbb{R}^{N \times H \times M \times L \times R}$, where $H$ denotes the number of heads.

**Parameterization in Position Contextualization.** The shapes of $\boldsymbol{W}_1$ and $\boldsymbol{W}_2$ allow for considerable flexibility. Given $\boldsymbol{e}_i \in \mathbb{R}^{H \times M \times L \times R}$. To achieve maximal expressiveness, $\boldsymbol{e}_i$ can be flattened into $\mathbb{R}^{HML \times R}$, with $\boldsymbol{W}_1$ and $\boldsymbol{W}_2 \in \mathbb{R}^{HML \times I}$. Alternatively, to minimize parameter usage, we set $\boldsymbol{W}_1$ and $\boldsymbol{W}_2$ as $\mathbb{R}^{H \times I}$, with weights shared across the $M$ and $L$ dimensions. $\psi$ is implemented through a standard MLP and the $I$ dimension is set to $4H$ in all experiments.

**Visualization of Tensor Operations.** To provide a clearer understanding of TAPE and the operation within the attention and feed-forward layers, we visualize the process in Fig. 4.

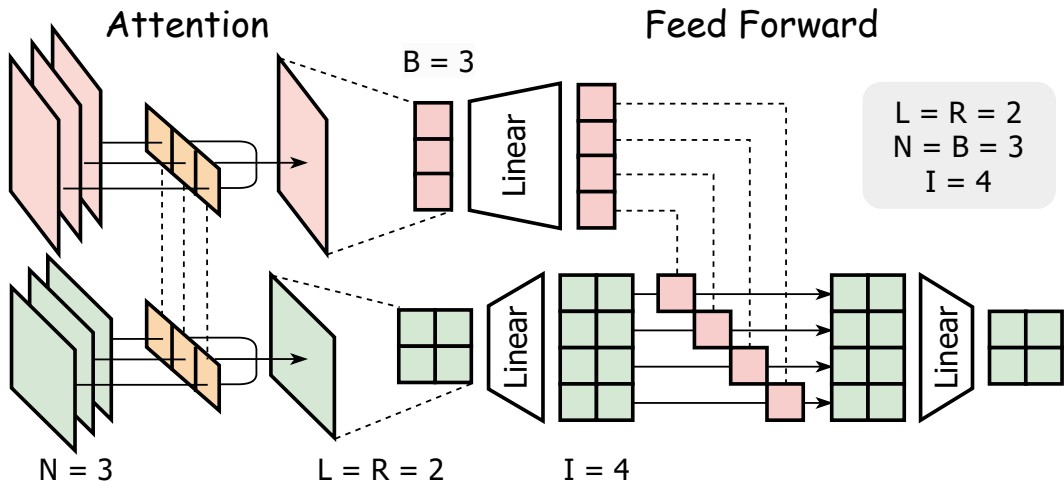

*Figure 4.* Visualization of TAPE's operations. The channel dimension is omitted for simplicity as all operations can be channel-wise. In the attention layer, the input token embeddings have a shape of $N \times B$, and the positional embeddings have a shape of $N \times L \times R$. For the feed-forward layer, the $N$ dimension is omitted as its operations are position-wise. The input token embeddings then have a shape of $B$ (or $B \times 1$), and the positional embeddings have a shape of $L \times R$.

## D. Additional Experiments

**Ablation Study on Architecture.** We ablate our architecture design for both attention layer and MLP layer in position contextualization. We conduct ablation studies on our architectural design for both the attention layer and the MLP layer in position contextualization. Additionally, we ablate two aspects of the design: rotation equivariance, by setting $\boldsymbol{W}_1, \boldsymbol{W}_2 \in \mathbb{R}^{HR \times I}$, which disrupts the $O(R)$-equivariance; the use of tensorial embeddings, by flattening $L = R = 2$ into $L = 1$ and $R = 4$; and both properties simultaneously, by setting $L = 4$ and $R = 1$. We use the same pre-training setting as Sec. 5.2 and directly report its perplexity in the test dataset of Github following He et al. (2024).

*Table 6.* Ablation study on TAPE architecture. We evalute pre-trained models' perplexity across varying sequence lengths on the GitHub test set.

| Architecture | | Perplexity | | | |
|---|---|---|---|---|---|
| **Attention** | **Feed Forward** | **128** | **256** | **512** | **1024** |
| ✗ | ✗ | 139.2 | 92.8 | 69.3 | 57.2 |
| ✗ | ✓ | 143.3 | 95.0 | 70.7 | 58.4 |
| ✓ | ✗ | 142.7 | 94.3 | 70.1 | 57.6 |
| ✓ | ✓ | 132.0 | 86.6 | 63.9 | 52.2 |
| **Rotation Equivariance** | **Tensorial Embedding** | | | | |
| ✗ | ✗ | 140.7 | 92.1 | 68.2 | 56.2 |
| ✓ | ✗ | 138.4 | 91.3 | 67.8 | 55.7 |
| ✗ | ✓ | 132.9 | 87.8 | 65.4 | 54.1 |
| ✓ | ✓ | 132.0 | 86.6 | 63.9 | 52.2 |

As shown in Tab. 6 , incorporating position contextualization in both the attention layer and the MLP layer results in the lowest perplexity across different positions within the training sequence length. Removing position contextualization from

either layer increases perplexity, even exceeding that of the traditional positional embedding without any architectural modifications. This outcome is reasonable, as applying position contextualization to only one component introduces an architectural inconsistency. Furthermore, ablating rotation equivariance allows all neurons in the positional embedding to undergo linear transformations, increasing the number of parameters but leading to worse results compared to TAPE. Similarly, reducing the tensorial embedding to a vector embedding leads to higher perplexities and a decline in performance.

**Ablation Study on TAPE Hyperparameter.** We aim to investigate the impact of varying $I$ on learning performance. Using the same pre-training settings as described in Section 4.2, we directly report the perplexity on the GitHub test dataset. As shown in Tab. 7, there is no significant difference when using different values of $I$, although a trend of first decreasing and then increasing can be observed. This suggests that a range of $I$ values from $2H = 24$ to $3H = 48$ may yield better performance compared to other settings. Therefore, as a general guideline, we recommend considering $I \in \{2, 3, 4\}H$ to optimize TAPE's performance.

*Table 7.* Ablation study on TAPE hyperparameter $I$. We evelute pre-trained models' perplexity across varying sequence lengths on the GitHub test set.

| TAPE | | Perplexity | | | |
|---|---|---|---|---|---|
| **Added Params. (M)** | **I** | **128** | **256** | **512** | **1024** |
| 0.11 | 12 | 133.2 | 87.9 | 65.2 | 53.6 |
| 0.22 | 24 | 133.0 | 86.1 | 63.2 | 51.8 |
| 0.44 | 48 | 132.0 | 86.6 | 63.9 | 52.2 |
| 0.88 | 96 | 133.2 | 87.5 | 64.5 | 52.7 |
| 1.76 | 192 | 133.0 | 87.3 | 64.5 | 53.0 |

**Stability of TAPE under Positional Shifts.** Stability in this context refers to the consistency of a sequence's representation under positional shifts (Sun et al., 2022). To evaluate the stability of TAPE, we examine two types of positional shifts: (1) appending [BOS] tokens at the beginning of the sequence and (2) initializing positional indices with non-zero values to simulate a phase shift (Sinha et al., 2022). We analyze two aspects of the representation: the attention weights and the dot product of positional embeddings, quantifying their changes after applying positional shifts. For comparison, we include RoPE, which also exhibits $O(R)$-equivariance ($R = 2$) and remains consistent across layers, as well as TAPE without equivariance, as explored in previous ablations.

*Table 8.* Comparison of RoPE, TAPE, and TAPE without equivariance (w/o EQ) under positional shifts. The table shows differences in attention weights (top) and positional embedding dot products (bottom) across layers for two shift methods: adding three [BOS] tokens ("Add Tokens") and starting position IDs at 3 ("Shift IDs").

| Atten. Diff. $(\times 10^{-2})$ | Add Tokens | | | | Shift IDs | | | |
|---|---|---|---|---|---|---|---|---|
| | Layer 1 | Layer 2 | Layer 4 | Layer 8 | Layer 1 | Layer 2 | Layer 4 | Layer 8 |
| RoPE | 8.93 | 8.51 | 12.29 | 11.46 | 0.01 | 0.02 | 0.02 | 0.03 |
| TAPE | 9.08 | 11.24 | 12.23 | 13.78 | 0.01 | 0.02 | 0.04 | 0.04 |
| w/o EQ | 11.30 | 11.38 | 13.32 | 14.55 | 0.01 | 0.24 | 0.37 | 0.51 |

| PE Dot Prod. Diff. (%) | Add Tokens | | | | Shift IDs | | | |
|---|---|---|---|---|---|---|---|---|
| | Layer 1 | Layer 2 | Layer 4 | Layer 8 | Layer 1 | Layer 2 | Layer 4 | Layer 8 |
| RoPE | 0.03 | 0.03 | 0.03 | 0.03 | 0.03 | 0.03 | 0.03 | 0.03 |
| TAPE | 0.03 | 0.37 | 2.75 | 6.62 | 0.03 | 0.02 | 0.03 | 0.04 |
| w/o EQ | 0.03 | 2.29 | 3.34 | 6.37 | 0.03 | 0.54 | 0.44 | 0.86 |

As shown in Tab. 8, TAPE demonstrates stability comparable to RoPE, maintaining consistent attention weights and positional embedding dot products across different layers. Among these, the near-zero change (not exactly zero, attributable

to numerical error observed in RoPE as well) in the dot-product when shifting IDs serves as empirical evidence for Proposition 4.2. However, when equivariance is removed from TAPE, the differences increase significantly, especially in deeper layers, highlighting the importance of equivariance in preserving stability.

**Additional Evaluation on Fine-tuned Llama-7b.** Modern benchmarks provide a comprehensive means to assess large language models' advanced capabilities in language understanding and reasoning. Accordingly, we further evaluate our fine-tuned Llama-7b (Sec. 5.3) on standard benchmarks, including ARC (Clark et al., 2018) and MMLU (Hendrycks et al., 2021).

*Table 9.* Accuracy in Percentage Across Methods and Benchmarks

| Method | MMLU (%) | | | | ARC (%) | |
|---|---|---|---|---|---|---|
| | **Humanities** | **Social Sciences** | **STEM** | **Other** | **Challenge** | **Easy** |
| LoRA | 39.09 ± 0.69 | 46.47 ± 0.88 | 33.65 ± 0.83 | 45.83 ± 0.89 | 45.31 ± 1.45 | 74.28 ± 0.90 |
| LongLoRA | 37.53 ± 0.69 | 43.55 ± 0.88 | 32.54 ± 0.83 | 43.84 ± 0.88 | 45.31 ± 1.45 | 74.16 ± 0.90 |
| ThetaScaling | 37.45 ± 0.69 | 43.16 ± 0.88 | 33.05 ± 0.83 | 44.64 ± 0.88 | 45.65 ± 1.46 | 74.24 ± 0.90 |
| TAPE | 37.96 ± 0.69 | 45.40 ± 0.88 | 33.27 ± 0.83 | 45.06 ± 0.88 | 46.25 ± 1.46 | 74.16 ± 0.90 |

As Tab. 9 shows, TAPE demonstrates notable performance compared to other methods on MMLU and ARC benchmarks. While TAPE's accuracy on MMLU is slightly lower than that of LoRA, it consistently outperforms others. On the ARC benchmark, TAPE performs comparably to other methods on the "Easy" subset but exhibits an advantage on the "Challenge" subset, underscoring its potential in complex reasoning tasks. Remarkably, these results are achieved using only fine-tuning, without pretraining TAPE, despite the presence of a certain degree of architectural shift.

**Additional Evaluation in Arithmetic Learning** We also evaluate the effectiveness of TAPE in Sec. 5.1 using a different training and testing length: 20/40 instead of 40/80. This setup is easier for the model to learn, with convergence achieved in less than half the steps. As shown in Fig. 5, TAPE outperforms FIRE with a marginal improvement of 5%. However, this improvement is less pronounced compared to the case with a train/test length of 40/80, suggesting that TAPE may be more effective in tackling complex and challenging tasks than simpler ones.

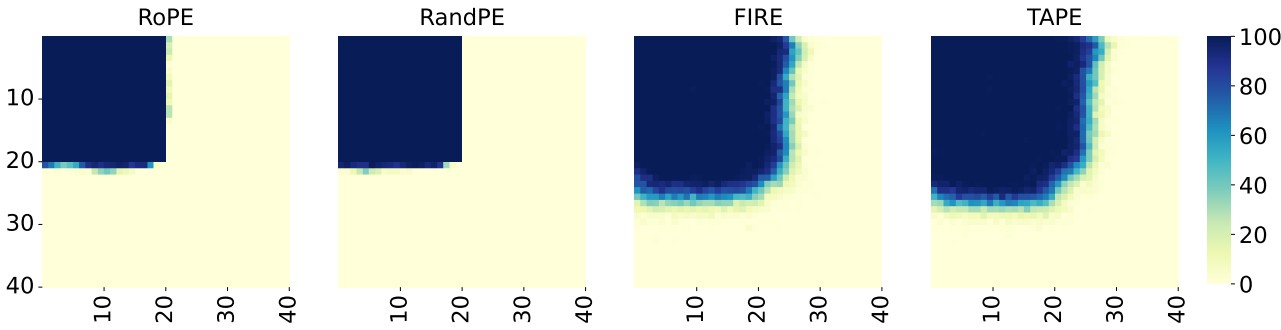

*Figure 5.* Accuracy on addition task trained with length 20 test on 2× context length. The average accuracy across the heatmap is 26.12%, 26.12%, 39.44% and 41.42% respectively for RoPE, RandPE, FIRE and TAPE.

**Integration with Extrapolation Technique.** Inspired by the demonstrated potential of NTK-based methods (Peng et al., 2023b) to enhance the length extrapolation ability of RoPE, we have explored integrating TAPE with such techniques when initialized as RoPE. Specifically, we selected the most recent method, YaRN (Peng et al., 2023b), and implemented its integration with TAPE to evaluate its performance in length extrapolation. The experiments were conducted under the same settings as described in Sec. 5.1. As shown in Fig. 6, the diagonal region exhibits darker colors, indicating higher accuracies. Quantitatively, YaRN effectively enhances the length extrapolation performance of TAPE with RoPE initialization, achieving a modest relative improvement of 3.4%. However, it still struggles to generalize to unseen sequences with significantly longer digit lengths.

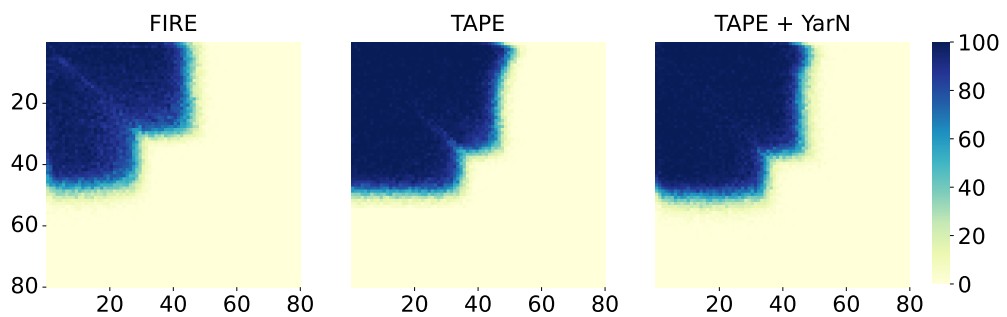

Figure 6. Accuracy on addition task on 2× context length. The average accuracy is 26.98%, 32.82% and 33.92% respectively for FIRE, TAPE and TAPE + YaRN.

Table 10. Comparing answers of different methods on example questions in QuALITY.

| Method | Question A | | Question B | |
|---|---|---|---|---|
| | Answer | EM | Answer | EM |
| Ground Truth | The secret service budget was small | ✓ | Only the private quarters or the office restroom | ✓ |
| TAPE | The secret service budget was small | ✓ | Only the private quarters | ✗ |
| xPos | They were all they were waiting for | ✗ | Only a tiny part of the right of the right to leave foreverish | ✗ |
| RandPE | Their human opinion was trusted by others who have trust the services of their people | ✗ | Only a handsome man | ✗ |
| RoPE | Their orless them together with their repories did not only they didn's never done was never done was never done... (repeating) | ✗ | The/O only the full-College All of the full-College All of the full-College... (repeating) | ✗ |
| ALiBi | Jimmy Carter is the president's de facto president | ✗ | Jimmy Carter is the president's de facto president | ✗ |

# E. Illustrations and Interpretation

**Visualization of PE Patterns.**    To better understand the impact of TAPE, we analyze its attention and positional embedding (PE) dot-product patterns. Fig. 7 compares the patterns of TAPE and RoPE in the last layer, while Fig. 8 illustrates the evolution of TAPE's dot-product patterns from shallow to deeper layers. The x-axis and y-axis correspond to the token positions of a sampled input sequence.

As shown in Fig. 7, TAPE demonstrates more evenly distributed long-range attention patterns, whereas RoPE tends to emphasize token locality. In Fig. 8, TAPE behaves similarly to RoPE in the first layer but gradually reduces the dominance of diagonal patterns as the depth increases. This transition results in the formation of grid-like patterns, indicating that the model starts to focus on distant tokens in a structured and periodic manner.

**Examples on QuALITY.**    To further validate TAPE's superior performance on the SCROLLS benchmark, we present two example questions from the QuALITY dataset within the SCROLLS benchmark. As shown in Tab. 10 and the detailed questions in Tab. 11, TAPE consistently generates either the correct answer or a response similar to the correct answer, even if not an exact match. In contrast, xPos and RandPE produce meaningful sentences that are unrelated to the specific question. RoPE and ALiBi, however, generate incoherent outputs: RoPE tends to repeat certain phrases, while ALiBi fails to recognize the presence of a question, producing the same irrelevant answer regardless of the input.

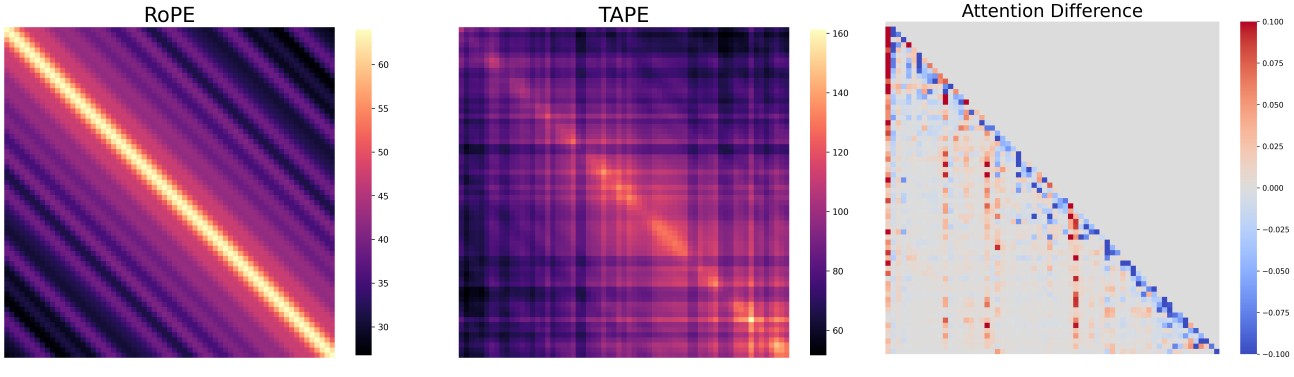

((a) ) Dot-product patterns of positional embeddings of TAPE and RoPE.      ((b) ) Difference between TAPE and RoPE

*Figure 7.* Comparison of TAPE and RoPE methods in terms of positional embedding dot-product patterns and their resulting attention differences. (a) TAPE demonstrates a systematic attention to surrounding tokens with relatively small dynamic ranges, whereas RoPE exhibits a highly significant diagonal pattern with distinctively black regions. (b) TAPE effectively attends to longer-range tokens, avoiding excessive attention to the self-token, in contrast to RoPE.

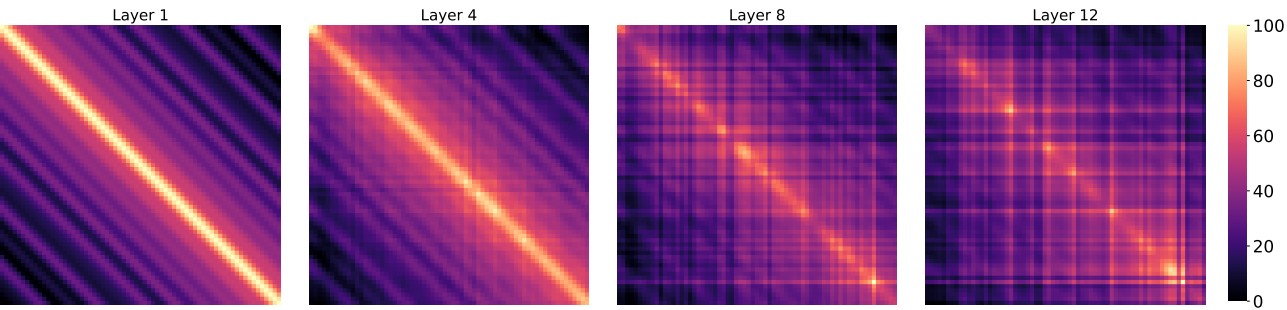

*Figure 8.* Dot-product patterns of positional embeddings in layers 1, 4, 8, and 12 (last) of TAPE.

Table 11: Example Questions in QuALITY

| **Qu. A** (ID: 20007_RZDMZJYW_2) | **Qu. B** (ID: 20007_RZDMZJYW_4) |
|---|---|
| What made it easier for previous presidents to get away with adultery? | Where in the White House is it feasible for the president to meet a woman? |
| (A) Their staff did not know
(B) They always tried to hide it well
(C) The secret service budget was small
(D) The reporters never found out | (A) Only the East Wing
(B) Only the private quarters
(C) Only the oval office, bowling alley, or East Wing
(D) Only the private quarters or the office restroom |

*Continued on next page...*

**Article Content:**

The logistics of presidential adultery.

The Washington Times could hardly contain its excitement: "A former FBI agent assigned to the White House describes in a new book how President Clinton slips past his Secret Service detail in the dead of night, hides under a blanket in the back of a dark-colored sedan, and trysts with a woman, possibly a celebrity, at the JW Marriott Hotel in downtown Washington." For Clinton-haters, Gary Aldrich's tale sounded too good to be true. And it was.

The not-so-Secret-Service agent's "source" turned out to be a thirdhand rumor passed on by Clinton scandalmonger David Brock. Those who know about White House security—Clinton staffers, the Secret Service, former aides to Presidents Reagan and Bush—demolished Aldrich's claims. Clinton couldn't give his Secret Service agents the slip (they shadow him when he walks around the White House), couldn't arrange a private visit without tipping off hotel staff, and couldn't re-enter the White House without getting nabbed. (Guards check all cars at the gate—especially those that arrive at 4 a.m.)

Even so, the image resonates. For some Americans, it is an article of faith: Bill Clinton cheated on his wife when he was governor, and he cheats on her as president. But can he? Is it possible for the president of the United States to commit adultery and get away with it? Maybe, but it's tougher than you think.

Historically, presidential adultery is common. Warren Harding cavorted with Nan Britton and Carrie Phillips. Franklin Roosevelt "entertained" Lucy Rutherford at the White House when Eleanor was away. America was none the wiser, even if White House reporters were.

Those who know Clinton is cheating often point to the model of John F. Kennedy, who turned presidential hanky-panky into a science. Kennedy invited mistresses to the White House for afternoon (and evening, and overnight) liaisons. Kennedy seduced women on the White House staff (including, it seems, Jackie's own press

*Continued on next page...*

secretary). Kennedy made assignations outside the White House, then escaped his Secret Service detail by scaling walls and ducking out back doors. If Kennedy did it, so can Clinton.

Well, no. Though Clinton slavishly emulates JFK in every other way, he'd be a fool to steal Kennedy's MO d'amour. Here's why:

1) Too many people would know. Kennedy hardly bothered to hide his conquests. According to Kennedy mistress (and mob moll) Judith Campbell's autobiography, those who knew about their affair included: Kennedy's personal aides and secretary (who pandered for him), White House drivers, White House gate guards, White House Secret Service agents, White House domestic staff, most of Campbell's friends, a lot of Kennedy's friends, and several Kennedy family members. Such broad circulation would be disastrous today because:

2) The press would report it. Kennedy conducted his affairs brazenly because he trusted reporters not to write about them. White House journalists knew about, or at least strongly suspected, Kennedy's infidelity, but never published a story about it. Ask Gary Hart if reporters would exercise the same restraint today. Clinton must worry about this more than most presidents. Not only are newspapers and magazines willing to publish an adultery story about him, but many are pursuing it.

For the same reason, Clinton would find it difficult to hire a mistress. A lovely young secretary would set off alarm bells in any reporter investigating presidential misbehavior. Says a former Clinton aide, "There has been a real tendency to have no good-looking women on the staff in order to protect him."

3) Clinton cannot avoid Secret Service protection. During the Kennedy era, the Secret Service employed fewer than 500 people and had an annual budget of about $4 million. Then came Lee Harvey Oswald, Squeaky Fromme, and John Hinckley. Now the Secret Service payroll tops 4,500 (most of them agents), and the annual budget exceeds $500 million (up 300 percent just since 1980). At any given time, more than 100 agents guard the president in the White House. Top aides from recent administrations are adamant: The Secret Service never lets the president escape its protection.

So what's a randy president to do? Any modern presidential affair would need to meet stringent demands. Only a tiny number of trusted aides and Secret Service agents could know of it. They would need to maintain complete silence about it. And no reporters could catch wind of it. Such an affair is improbable, but—take heart, Clinton-haters—it's not impossible. Based on scuttlebutt and speculation from insiders at the Clinton, Bush, Reagan, and Ford White Houses, here are the four likeliest scenarios for presidential adultery. 1) The White House Sneak. This is a discreet variation of the old Kennedy/Campbell liaison. It's late at night. The president's personal aides have gone home. The family is away. He is alone in the private quarters. The private quarters, a.k.a. "the residence," occupy the second and third floors of the White House. Secret Service agents guard the residence's entrances on the first floor and ground floors, but the first family has privacy in the quarters themselves. Maids and butlers serve the family there, but the president and first lady ask them to leave when they want to be alone. The president dials a "friend" on his private line. (Most presidents placed all their calls through the White House operators, who kept a record of each one; the Clintons installed a direct-dial line in the private quarters.) The president invites the friend over for a cozy evening at the White House. After he hangs up with the friend, he phones the guard at the East Executive Avenue gate and tells him to admit a visitor. He also notifies the Secret Service agent and the usher on duty downstairs that they should send her up to the residence.

A taxi drops the woman near the East gate. She identifies herself to the guard, who examines her ID, runs her name through a computer (to check for outstanding warrants), and logs her in a database. A White House usher escorts her into the East Wing of the White House. They walk through the East Wing and pass the Secret Service guard post by the White House movie theater. The agent on duty waves them on. The usher takes her to the private elevator, where another Secret Service agent is posted. She takes the elevator to the second floor. The president opens the door and welcomes her. Under no circumstances could she enter the living quarters without first encountering Secret Service agents.

Let us pause for a moment to demolish two of the splashier rumors about White House fornication. First, the residence is the only place in the White House where the president can have safe (i.e., uninterrupted) sex. He can be intruded upon or observed everywhere else—except, perhaps, the Oval Office bathroom. Unless the president is an exhibitionist or a lunatic, liaisons in the Oval Office, bowling alley, or East Wing are unimaginable. Second, the much-touted tunnel between the White House and the Treasury Department is all-but-useless to the presidential adulterer. It is too well-guarded. The president could smuggle a mistress through it, but it would attract far more attention from White House staff than a straightforward gate entry would.

Meanwhile, back in the private quarters, the president and friend get comfortable in one of the 14 bedrooms (or, perhaps, the billiard room). After a pleasant 15 minutes (or two hours?), she says goodbye. Depending on how long she stays, she may pass a different shift of Secret Service agents as she departs. She exits the White House grounds, unescorted and unbothered, at the East gate.

The Risks: A gate guard, an usher, and a handful of Secret Service agents see her. All of them have a very good idea of why she was there. The White House maid who changes the sheets sees other suspicious evidence. And the woman's—real—name is entered in a Secret Service computer. None of this endangers the president too much. The computer record of her visit is private, at least for several decades after he leaves office. No personal aides know about the visit. Unless they were staking out the East gate, no journalists do either. The Secret Service agents, the guard, the steward, and the maid owe their jobs to their discretion. Leaks get them fired.

That said, the current president has every reason not to trust his Secret Service detail. No one seriously compares Secret Service agents (who are pros) to Arkansas state troopers (who aren't). But Clinton might not trust any

security guards after the beating he took from his Arkansas posse. Also, if other Secret Service agents are anything like Aldrich, they may dislike this president. One Secret Service leak—the lamp-throwing story—already damaged Clinton. Agents could tattle again.

2) The "Off-the-Record" Visit. Late at night, after his personal aides and the press have gone home, the president tells his Secret Service detail that he needs to take an "off-the-record" trip. He wants to leave the White House without his motorcade and without informing the press. He requests two agents and an unobtrusive sedan. The Secret Service shift leader grumbles but accepts the conditions. Theoretically, the president could refuse all Secret Service protection, but it would be far more trouble than it's worth. He would have to inform the head of the Secret Service and the secretary of the Treasury.

The president and the two agents drive the unmarked car to a woman friend's house. Ideally, she has a covered garage. (An apartment building or a hotel would raise considerably the risk of getting caught.) The agents guard the outside of the house while the president and his friend do their thing. Then the agents chauffeur the president back to the White House, re-entering through the Southwest or Southeast gate, away from the press station.

The Risks: Only two Secret Service agents and their immediate supervisor know about the visit. It is recorded in the Secret Service log, which is not made public during the administration's tenure. Gate guards may suspect something fishy when they see the car. A reporter or passer-by could spy the president—even through tinted windows—as the car enters and exits the White House. The friend's neighbors might spot him, or they might notice the agents lurking outside her house. A neighbor might call the police to report the suspicious visitors. All in all, a risky, though not unthinkable, venture.

3) The Camp David Assignation. A bucolic, safer version of the White House Sneak. The president invites a group of friends and staffers—including his paramour but not his wife—to spend the weekend at Camp David. The girlfriend is assigned the cabin next to the president's lodge. Late at night, after the Hearts game has ended and everyone has retired to their cabins, she strolls next door. There is a Secret Service command post outside the cabin. The agents on duty (probably three of them) let her enter. A few hours later, she slips back to her own cabin.

The Risks: Only a few Secret Service agents know about the liaison. Even though the guest list is not public, all the Navy and Marine personnel at Camp David, as well as the other guests, would know that the presidential entourage included an attractive woman, but not the first lady. That would raise eyebrows if it got back to the White House press room.

4) The Hotel Shuffle. The cleverest strategy, and the only one that cuts out the Secret Service. The president is traveling without his family. The Secret Service secures an entire hotel floor, reserving elevators and guarding the entrance to the president's suite. The president's personal aide (a man in his late 20s) takes the room adjoining the president's. An internal door connects the two rooms, so the aide can enter the president's room without alerting the agents in the hall. This is standard practice. Late in the evening, the aide escorts a comely young woman back to the hotel. The

Secret Service checks her, then waves her into the aide's room. She emerges three hours later, slightly disheveled. She kisses the aide in the hall as she leaves. Someone got lucky—but who?

The Risks: The posted Secret Service agents might see through the charade. More awkwardly, the aide would be forced to play the seamy role of procurer. (He would probably do it. Kennedy's assistants performed this task dutifully.)

In short, presidential adultery is just barely possible in 1996. But it would be extremely inconvenient, extremely risky, and potentially disastrous. It seems, in fact, a lot more trouble than it's worth. A president these days might be wiser to imitate Jimmy Carter, not Jack Kennedy, and only lust in his heart.

