# OpenReview forum: "Rethinking Addressing in Language Models via Contextualized Equivariant Positional Encoding"
_ICML.cc/2025/Conference — ICML 2025 poster_

### Official Review · Reviewer_udZG · 2025-03-13

**Overall Recommendation:** 3

**Summary:**

This paper proposes a new positional encoding method for LLMs, which could enhance the position-addressing ability of transformers. Permutation and orthogonal equivariance are also applied to enforce the positional encoding. This method demonstrates superior performance on various tasks, especially long-context tasks, such as passkey retrieval tasks.

**Claims And Evidence:**

the claims are supported by experiments results and theoretical proof.

**Essential References Not Discussed:**

N/A

**Experimental Designs Or Analyses:**

I have checked the author’s experimental results and the baselines used for comparison. Regarding the need for more comparative methods on long-context tasks, I will elaborate on this specifically in the weakness.

**Methods And Evaluation Criteria:**

the authors propose context-aware positional encodings aiming to improve positional encoding and enhance the performance of LLMs during both pre-training and fine-tuning.

**Other Comments Or Suggestions:**

Several typos need to be corrected :
1. “superioroty” should be “superiority” in Line 76.
2. “fine-fining”should be “fine-tuning” in Line 399.

**Other Strengths And Weaknesses:**

Strengths: The authors propose a dynamic, context-aware positional encoding, which can be applied during both the pre-training and fine-tuning stages. They provide extensive experimental results in the main paper and appendix, demonstrating the effectiveness of the method across various tasks. Additionally, the paper is well-structured, making it easy to understand.

Weakness: Existing positional encoding methods have introduced improvements to RoPE to better adapt it for long-sequence tasks, such as NTK-aware RoPE [4]. Could you provide comparative experimental results of this method on long-context tasks?

[4] Peng, Bowen, et al. "Yarn: Efficient context window extension of large language models." 2023.

**Questions For Authors:**

Computing a positional embedding at each layer increases the computational cost for larger-scale language models. Is the added time acceptable, and does the proposed method have scalability? Please provide a time complexity analysis.

**Relation To Broader Scientific Literature:**

This paper contributes to the broader scientific literature by introducing a novel context-aware positional encoding method for LLMs, enhancing their position-addressing ability through permutation and orthogonal equivariance. It builds upon prior work on positional encoding techniques like RoPE[1], ALiBi[2].

[1] Su, Jianlin, et al. "Roformer: Enhanced transformer with rotary position embedding." Neurocomputing, 2024

[2]Press, Ofir, Noah A. Smith, and Mike Lewis. "Train short, test long: Attention with linear biases enables input length extrapolation.” 2021.

**Theoretical Claims:**

I checked the proof of the proposition, and I think no issues with it.

---

> ### Author Rebuttal · Authors · 2025-04-01
>
> We greatly thank Reviewer udZG for appreciating our contributions, providing valuable suggestions on improving the work, and supporting the acceptance of this work. We address the questions as follows.
>
> > W1: Existing positional encoding methods have introduced improvements to RoPE to better adapt it for long-sequence tasks, such as NTK-aware RoPE [4]. Could you provide comparative experimental results of this method on long-context tasks?
>
> Thank you for your suggestion. We have implemented YaRN as an additional baseline in our SCROLLS benchmark experiments. The training is currently in progress, and we will update the results once they are available.
>
> > Q1: Computing a positional embedding at each layer increases the computational cost for larger-scale language models. Is the added time acceptable, and does the proposed method have scalability? Please provide a time complexity analysis.
>
> As shown in Table 3, TAPE introduces <1% additional parameters and approximately 12% increased computational cost (FLOPs). Since scaling typically involves stacking these layers, the overhead remains proportional. We believe the modest increase in memory usage and ~10% additional compute time is a reasonable trade-off for the benefits provided.

---

### Official Review · Reviewer_poAq · 2025-03-13

**Overall Recommendation:** 3

**Summary:**

This paper introduces TAPE ,a novel approach to enhancing position-based addressing in Transformers by dynamically adapting positional encodings across layers based on sequence context. TAPE ensures stability and robustness by enforcing permutation and orthogonal equivariance. Experimental results demonstrate that TAPE outperforms existing positional encoding techniques in language modeling, arithmetic reasoning, and long-context retrieval tasks.

**Claims And Evidence:**

Yes, the claims are.

**Essential References Not Discussed:**

No, there aren’t.

**Experimental Designs Or Analyses:**

Yes, I checked the experimental results tables and their corresponding analysis.

**Methods And Evaluation Criteria:**

Yes, the proposed methods do.

**Other Comments Or Suggestions:**

- Figure 3 could be represented using a different type of chart and would look more visually appealing if it occupies half a column.

**Other Strengths And Weaknesses:**

Strengths：

- The proposed method is novel and effective, with corresponding theoretical support.

- The extensive experimental setup, especially in terms of performance on long-context tasks provides strong evidence of the effectiveness of the proposed method.

Weaknesses：

- The authors provide the running time of attention layers as experimental results in Table 4. However, since the proposed method updates positional features layer-wise through interactions and joint training with token representations in every Transformer layer, what is the experimental runtime during training and inference for large-scale language models?

- Since the proposed method updates positional features layer-wise, how does this affect gradient in the model? Could the authors provide an analysis or empirical results on gradient changes to better understand the impact of this method on training process?

**Questions For Authors:**

- In Table 2 and Table 4, the authors present experimental results showing that the proposed method performs well on long-context tasks. I am curious whether this strong performance in long-context scenarios has theoretical support.

**Relation To Broader Scientific Literature:**

The Positional Encoding proposed in this paper aims to unleash the power of position-based addressing, as existing methods have gradually weakened this capability[1,2].

[1]Roformer: Enhanced transformer with rotary position embedding. Neurocomputing,2024

[2]Sun, Yutao, et al. "A length-extrapolatable transformer." 2022

**Theoretical Claims:**

Yes, I checked the formulas in Section 3.

---

> ### Author Rebuttal · Authors · 2025-04-01
>
> We greatly thank Reviewer poAq for appreciating our contributions, providing valuable suggestions on improving the work, and supporting the acceptance of this work. We address the questions as follows.
>
> >W1: The authors provide the running time of attention layers as experimental results in Table 4. However, since the proposed method updates positional features layer-wise through interactions and joint training with token representations in every Transformer layer, what is the experimental runtime during training and inference for large-scale language models?
>
> Thank you for your suggestion. We have included the experimental runtime for full model inference in the updated table below.
> | Method | TAPE | RoPE | FIRE | T5’s relative bias |
> | --- | --- | --- | --- | --- |
> | Samples Per Second | 58.6 | 71.8 | 33.4 | 46.8 |
>
> >W2: Since the proposed method updates positional features layer-wise, how does this affect gradient in the model? Could the authors provide an analysis or empirical results on gradient changes to better understand the impact of this method on training process?
>
> We appreciate this insightful question. During our experiments, we monitored gradient norms and did not observe significant differences compared to RoPE. However, as this aspect falls outside our primary focus (motivation, methodology, and significance), we did not conduct an in-depth analysis of gradient behavior.
>
> >Q1: In Table 2 and Table 4, the authors present experimental results showing that the proposed method performs well on long-context tasks. I am curious whether this strong performance in long-context scenarios has theoretical support.
>
> TAPE's superior performance in long-context scenarios can be attributed to two key factors: First, its learnable nature enables dynamic adaptation to varying context lengths. Second, as formally established in Proposition 3.1, our relative position encoding scheme possesses inherent generalization capabilities to unseen sequence lengths.
> In contrast, conventional positional encoding methods exhibit fundamental limitations in long-context settings, as they either rely on predetermined distance-decay patterns, or lack this crucial relativity property.

---

### Official Review · Reviewer_G28F · 2025-03-14

**Overall Recommendation:** 4

**Summary:**

This paper introduces a new approach to processing language sequences using transformer blocks, where token features and positional embeddings are combined and contextualized. The authors extend traditional positional encoding by dividing it into multiple blocks, allowing for more flexible associations between tokens and their positions. They also ensure that their functions for token mixing and position contextualization are equivariant to permutations and orthogonal transformations, addressing limitations in existing models. The overall goal is to improve how transformer models process both token features and positional information in sequences.

## update after rebuttal

I think this is a good paper and keep my positive score.

**Claims And Evidence:**

Yes.

**Essential References Not Discussed:**

None.

**Experimental Designs Or Analyses:**

I check all experiments.

**Methods And Evaluation Criteria:**

Yes.

**Other Comments Or Suggestions:**

None.

**Other Strengths And Weaknesses:**

Strengths
- Instead of a specialized architecture, this paper modifies attention and MLP layers of Transformer, allowing easy integration into existing transformers.
- Contextualized positional embeddings allow the model to dynamically adjust how positional information is interpreted depending on the surrounding tokens, improving its ability to capture more nuanced relationships between tokens.
- The authors conduct thorough evaluations on various tasks, including passkey retrieval, arithmetic learning, and training from scratch with different context lengths. Additionally, they provided clear and insightful visualizations of attention maps, which further enhance the understanding of their model's behavior.

Weaknesses
- I do not see any major weaknesses. One suggestion is about the study of hyper-parameters. I think the authors choose B, L, R to align with RoPE for comparison. Since the proposed embeddings are learnable, it would be interesting to see how those hyper-parameters could affect the performance.

**Questions For Authors:**

None.

**Relation To Broader Scientific Literature:**

It provides a new way to handle positions in Transformer, which can be wildly applied to diverse domains.

**Theoretical Claims:**

Yes, I check the proof of Proposition 3.1.

---

> ### Author Rebuttal · Authors · 2025-04-01
>
> We sincerely appreciate Reviewer G28F's positive assessment of our contributions and strong endorsement for acceptance. The reviewer provided one suggestion, to which we respond below:
>
> > I do not see any major weaknesses. One suggestion is about the study of hyper-parameters. I think the authors choose B, L, R to align with RoPE for comparison. Since the proposed embeddings are learnable, it would be interesting to see how those hyper-parameters could affect the performance.
>
> Thank you for your valuable suggestion. We have conducted hyperparameter exploration (detailed in the Appendix D), and our experiments demonstrate that the RoPE-like initialization yields the best performance among the configurations we tested.

---

### Official Review · Reviewer_qpJK · 2025-03-17

**Overall Recommendation:** 4

**Summary:**

This paper proposes a new method for learnable positional encodings, where they are allowed to depend on context/content. The positional encodings, termed TAPE (“conTextualized equivariAnt Position Encoding”), can be added to pre-trained transformers, with only the TAPE-relevant parameters fine-tuned. TAPE is permutation and orthogonal equivariant and uses higher order tensors, and performs well in experiments spanning arithmetic reasoning, long-context retrieval, and language modeling.

**Claims And Evidence:**

L051, “This rigidity [the fixed distance dependence / locality bias] limits the ability of positional encodings to model long-range dependencies and makes it challenging to attend to distant query-key pairs.” Is there a citation or experiment to support the claim that it is the positional encodings, in particular, that make long-range dependencies hard for LLMs?

L686: “To the best of our knowledge, we are the first to introduce equivariance in language models, recognizing the symmetry in positional embeddings.” This is a strong claim. Is this not already done by RoPE, as noted in the paper?

**Essential References Not Discussed:**

There are other references that deal explicitly with general group equivariance in positional encodings for geometric applications, but I don’t think they are essential.

**Experimental Designs Or Analyses:**

The authors explain the success of TAPE in terms of permutation and orthogonal equivariance, as well as the tenderized representations, but only ablate orthogonal equivariance — this is the only area I see for improvement, as the other experiments seem very thorough.

**Methods And Evaluation Criteria:**

Yes, although the highlighted benchmark datasets seem to be chosen as tasks where TAPE is expected to perform well (arithmetic and long context). However, other datasets are included in the appendix.

**Other Comments Or Suggestions:**

I would recommend making Proposition 3.1 more general, by explicitly stating the necessary assumptions on E rather than immediately specializing to RoPE and random Fourier features. Also, would the result of Prop 3.1 not hold if f and g satisfied (4) and (5) for only R in the set of permutation matrices, rather than orthogonal matrices? This counterfactual would be good to include as part of the statement, if Prop 3.1 is supposed to motivate orthogonal invariance specifically.

Figure 2: can the authors specify what the x and y-axis are in the figure caption?

As a minor comment, there were a lot of typos. Here are some of them:

Typos:
L99, “adiditionaly”
L77, “superioroty”
L150, “transformer” -> “transformers”
L162, “encoding” -> “encodings”
L237, “conTexturalized”
L246, “O(r)-invariance” —> “O(r)-invariant”
L285, “test” —> “tested”
L297: “arthimetic”
L663, “quardratic”
L672: “focus” -> “focuses”
L792, “Contextulization”

**Other Strengths And Weaknesses:**

Strengths:

The proposed positional encodings, TAPE, perform well compared to baselines on arithmetic and long-context tasks. They also admit an efficient implementation, and enable parameter-efficient fine-tuning that works better than LoRA and LongLoRA on passkey retrieval. Several additional experiments that I might have requested were already in the appendix, including ablations of orthogonal equivariance, and evaluation on other LLM tasks where long context is not necessarily the main challenge.

Weaknesses:

The motivation of the technique is a bit confusing. The authors claim that relative positional encodings are crucial for “stability and generalization to varying sequence lengths” (L223-224), and use relative positional encodings in their formulation (equation 6) and prove that the transformer is invariant to token index shift (Prop 3.1), but then highlight an arithmetic task where absolute positions are necessary. Also, the use of tensors is not very well-motivated. Overall, although the experimental performance is seemingly very good, the design of TAPE seems a bit ad-hoc. The paper also cites geometric learning as inspiration — “This approach is inspired from the studies for geometric deep learning which processes graphs and point clouds by integrating token features with their geometric properties while preserving inherent physical symmetries” (L83) — but isn’t this property (positional encodings depending on relative distances) already satisfied by RoPE? Some additional ablations that could help clarify the design of TAPE include ablating the tensorial nature (reducing dimensions), ablating the dependence on context in the positional embeddings, etc.

**Questions For Authors:**

1. The arithmetic task is one where absolute positions are necessary (L316), but TAPE uses relative positions (Prop 3.1) — how then does TAPE do well? (Clarification of TAPE’s advantages and understanding of experimental results)
2. Is there an indexing problem with equation 7? Don’t the two sums cancel each other out? (Clarification for assessing paper’s accuracy)
3. Do the same hyperparameter choices (dimensions, etc) work across different tasks? (Affects my evaluation of advantages/disadvantages of the method)

**Relation To Broader Scientific Literature:**

The paper presents a novel positional encoding, building on RoPE and others. It draws on insights from papers like Ebrahimi et al 2024 and Sinha et al 2022.

**Theoretical Claims:**

n/a

---

> ### Author Rebuttal · Authors · 2025-04-01
>
> We greatly thank Reviewer qpJK for appreciating our contributions. We address the concerns as follows.
>
> > W1: The motivation of the technique is a bit confusing. The authors claim that relative positional encodings are crucial for “stability and generalization to varying sequence lengths” (L223-224), and use relative positional encodings in their formulation (equation 6) and prove that the transformer is invariant to token index shift (Prop 3.1), but then highlight an arithmetic task where absolute positions are necessary.
>
> We apologize for the inaccuracy in our original statement (Line 316) regarding absolute positions that caused confusion. Please refer to our response to Q1 to see whether this clarification addresses the point.
>
> ---
> >W2: Also, the use of tensors is not very well-motivated. Overall, although the experimental performance is seemingly very good, the design of TAPE seems a bit ad-hoc.
>
> We acknowledge that certain implementation choices in TAPE, such as the tensorial embedding design, were empirically driven to optimize performance. As noted in Appendix D, our ablation studies validate the effectiveness of these architectural decisions.
>
> ---
> >W3: The paper also cites geometric learning as inspiration — “This approach is inspired from the studies for geometric deep learning which processes graphs and point clouds by integrating token features with their geometric properties while preserving inherent physical symmetries” (L83) — but isn’t this property (positional encodings depending on relative distances) already satisfied by RoPE? Some additional ablations that could help clarify the design of TAPE include ablating the tensorial nature (reducing dimensions), ablating the dependence on context in the positional embeddings, etc.
>
> Yes, RoPE also satisfies this property, which is precisely why we employ it as one of our instantiation methods. We do not claim this property as our novelty, but rather use it as a principle to motivate our design choices regarding contextualized position embeddings. As shown in Appendix D, we provide ablation studies that: (1) validate the design of equivariant and tensorial embeddings, (2) analyze the effects of Attention and MLP layers, and (3) investigate the impact of hyperparameter choices. Note that ablating the context-dependence in our positional embeddings (our core design) reduces to RoPE, which is consequently included as a baseline in our experiments.
>
> ---
> > Q1: The arithmetic task is one where absolute positions are necessary (L316), but TAPE uses relative positions (Prop 3.1) — how then does TAPE do well? (Clarification of TAPE’s advantages and understanding of experimental results)
>
> We appreciate this opportunity to clarify. There was an inaccuracy in the original L316 statement regarding absolute positions - we have now corrected this, as absolute positions are not necessarily required for arithmetic. The critical factor is learning the relative importance of different positions within the sequence. As detailed in L328 onwards, TAPE is able to learn these position-dependent importance relationships within the task context. A detailed explanation is also attached:
>
> In arithmetic tasks, every digit has equal importance to the equation, regardless of its distance from the output. Traditional positional embeddings often assume a distance-decay effect, where words farther apart are less significant in the output. While this assumption is valid for most language tasks, it does not hold for arithmetic tasks. Positional contextualization enables dynamic reweighting of positional importance based on the task context, preserving effective distance decay for language tasks while addressing arithmetic contexts appropriately. This highlights TAPE’s potential advantages in arithmetic tasks.
>
> ---
> > Q2: Is there an indexing problem with equation 7? Don’t the two sums cancel each other out? (Clarification for assessing paper’s accuracy)
>
> No, the two sums in Equation 7 form a linear combination of vectors with weights summing to 1, making they cannot cancel out.
>
> ---
> > Q3: Do the same hyperparameter choices (dimensions, etc) work across different tasks? (Affects my evaluation of advantages/disadvantages of the method)
>
> Yes, we maintain consistent hyperparameters across all tasks (with L=R=2 in our main experiments). Additionally, Appendix D provides detailed ablation studies analyzing the impact of different hyperparameter choices.
>
> ---
> *Response to Suggestions:*
> 1. Prop 3.1:  The orthogonality of R is indeed crucial for Proposition 3.1 to hold, as non-orthogonal transformations would violate the invariance properties demonstrated in Appendix B. While generalizing the assumptions is an interesting direction, we believe this extension merits dedicated future research.
> 2. Figure 2: The x- and y-axes represent the sequence lengths of the two operands respectively.
> 3. Typos: Thank you for your careful review. We have fixed all of them.

---

### Decision · Program_Chairs · 2025-05-01

**Decision:**

Accept (poster)

**Comment:**

The paper introduces TAPE, a method designed to enhance positional embeddings by incorporating sequence content across layers. TAPE can be integrated into pre-trained transformers, with only the relevant parameters fine-tuned. The proposed method is novel, and it demonstrates strong performance compared to baselines on arithmetic and long-context tasks. Extensive experiments are conducted, and the technology quality is solid. Overall, this is a strong paper. For the final version, the authors should clarify the motivation further and provide additional results as per the reviewers' comments.